# Developmental trajectory and evolutionary origin of thymic mimetic cells

Anja Nusser[1,5,7], Oliver S. Thomas[1,7], Gaoqun Zhang[1,2,7], Daisuke Nagakubo[1,6], Laura Arrigoni[3], Brigitte Krauth[1,2] & Thomas Boehm[1,2,4 ✉]

The generation of self-tolerant repertoires of T cells depends on the expression of peripheral self antigens in the thymic epithelium[1] and the presence of small populations of cells that mimic the diverse phenotypes of peripheral tissues[2–7]. Whereas the molecular underpinnings of self-antigen expression have been extensively studied[8], the developmental origins and differentiation pathways of thymic mimetic cells remain to be identified. Moreover, the histological identification of myoid and other peripheral cell types as components of the thymic microenvironment of many vertebrate species[9] raises questions regarding the evolutionary origin of this unique tolerance mechanism. Here we show that during mouse development, mimetic cells appear in the microenvironment in two successive waves. Cells that exhibit transcriptional signatures characteristic of muscle, ionocyte, goblet and ciliated cells emerge before birth, whereas others, such as those that mimic enterohepatic cells and skin keratinocytes, appear postnatally. These two groups also respond differently to modulations of thymic epithelial cell progenitor pools caused by deletions of *Foxn1* and *Ascl1*, expression of a hypomorphic variant of the transcription factor FOXN1, and overexpression of the signalling molecules BMP4 and FGF7. Differences in mimetic cell populations were also observed in thymic microenvironments reconstructed by replacement of mouse *Foxn1* with evolutionarily ancient Foxn1/4 gene family members, including the *Foxn4* gene of the cephalochordate amphioxus and the *Foxn4* and *Foxn1* genes of a cartilaginous fish. Whereas some cell types, such as ciliated cells, develop in the thymus in the absence of FOXN1, mimetic cells that appear postnatally, such as enterohepatic cells, require the activity of the vertebrate-specific transcription factor FOXN1. The thymus of cartilaginous fishes and the thymoid of lampreys, a representative of jawless vertebrates, which exhibit an alternative adaptive immune system[10], also harbour cells that express genes encoding peripheral tissue components such as the liver-specific protein transthyretin. Our findings suggest an evolutionary model of successive changes of thymic epithelial genetic networks enabling the coordinated contribution of peripheral antigen expression and mimetic cell formation to achieve central tolerance for vertebrate-specific innovations of tissues such as the liver[11,12].

The emergence of somatic diversification of antigen receptor genes required a radical re-organization of adaptive immune facilities to avoid the potentially fatal autoimmunity associated with quasi-random receptor specificities. It is conceivable, however, that the initial diversity of antigen receptor repertoires in early vertebrates was much lower than that of extant vertebrates[13], allowing a grace period during which appropriate tolerance mechanisms could have evolved in step with a gradually increasing diversity of antigen receptor repertoires. For instance, the emergence of new antigen presentation pathways and/or the repurposing of ancient antigen presentation pathways[14] is thought to have contributed to the diverse central and peripheral tolerance mechanisms that characterize the extant vertebrate immune systems. In the thymus, central tolerance is induced by at least two functionally connected mechanisms. The expression of peripheral genes by medullary epithelial cells[1] and the presence of peripheral mimetic cells[2–7] combine to provide developing T cells with a means to probe the

[1]Department of Developmental Immunology, Max Planck Institute of Immunobiology and Epigenetics, Freiburg, Germany. [2]Evolutionary Immunology Group, Max Planck Institute for Biology, Tübingen, Germany. [3]Deep Sequencing Facility, Max Planck Institute of Immunobiology and Epigenetics, Freiburg, Germany. [4]Institute for Immunodeficiency, Center for Chronic Immunodeficiency (CCI), University Medical Center, Faculty of Medicine, University of Freiburg, Freiburg, Germany. [5]Present address: Center for Personalized Medicine (ZPM), University Hospital Tübingen, Tübingen, Germany. [6]Present address: Department of Pharmaceutical Sciences, School of Pharmacy, International University of Health and Welfare, Ohtawara, Japan. [7]These authors contributed equally: Anja Nusser, Oliver S. Thomas, Gaoqun Zhang. ✉e-mail: boehm@ie-freiburg.mpg.de

specificity of their antigen receptors against the universe of peripheral self antigens[15]. Peripheral cell types, such as muscle-like cells, goblet and mucous cells, have long been known to be present in the thymic microenvironment of many different vertebrate species[9,16,17]. For instance, as early as 1905, Hammar concluded that myoid cells are peculiarly differentiated epithelial cells and therefore unrelated to true muscle cells[17], although their origin from extrathymic tissue sources has later also been considered[18]. A tolerance-inducing function of peripheral cell types in the thymic microenvironment was suggested more than 50 years ago[19], but this attribute of thymic mimetic cells has only recently been experimentally verified[2-7]. The presence of peripheral cell types in the thymus of all jawed vertebrates[9] prompted us to examine further aspects of their development and evolutionary origin.

## Development of mimetic TEC subsets

Focusing on major thymic mimetic cell types[2], we re-examined the cellular heterogeneity of EPCAM⁺CD45⁻ thymic epithelial cells (TECs) at various developmental time points, for which we had already identified two progenitor populations, in addition to cortical TECs (cTECs) and medullary TECs (mTECs)[20]. Signature gene sets for mimetic cell types were derived from single-cell differential gene expression data[2] and used as input to AUCell[21] to score each signature per cell (Extended Data Fig. 1a,b). Cells with signatures of early progenitors and cTECs dominate the epithelial compartment at embryonic day 16.5 (E16.5), whereas those characterizing postnatal progenitors and mTECs constitute the majority at postnatal day 28 (P28); at birth (P0), the epithelial populations have an intermediate composition (Extended Data Fig. 1c and Supplementary Table 1). The 11 mimetic cell types[2] examined here are robustly detected at P28 (Extended Data Fig. 1c). Whereas at P28 essentially all Aire-stage cells express the genes encoding the tolerogenic factors AIRE[22] and FEZF2[23], only about a quarter of mimetic cells express *Aire* (compatible with their post-Aire phenotype[2]); by contrast, more than 70% of mimetic cells express *Fezf2* (Extended Data Fig. 1d). At E16.5, mimetic cells are essentially undetectable, with the exception of a few muscle and goblet cells (Extended Data Fig. 1c). At P0, muscle cells represent a major fraction of mimetic cells; at this stage, *Myog*-expressing cells often occur in small medullary clusters (Extended Data Fig. 2a,b), whereas they appear to be more scattered at P28 (Extended Data Fig. 2c). Enumeration of *Myog*-expressing cells by RNA in situ hybridization (ISH) indicates that they are about three times more frequent at P0 than at P28 (Extended Data Fig. 2d), validating the single-cell RNA-sequencing (scRNA-seq) data (Extended Data Fig. 1c and Supplementary Table 1). Other mimetic cell types are also predominantly found in the medulla (Extended Data Fig. 2e,f). To quantify these changes among cell types during development, we performed differential abundance analysis of cellular compositions using scCODA[24] (Extended Data Fig. 1e). Compared with the P28 time point, TECs at E16.5 exhibit a greater abundance of early progenitors and cTECs, and fewer postnatal progenitors, mTECs, Aire-stage cells and tuft cells. At P0, we still found larger early progenitor and cTEC populations, and slightly smaller muscle populations (Extended Data Fig. 1e).

However, the sensitivity of differential abundance analysis for rare cell types (such as the thymic mimetic cells) is relatively low, complicating its use for comparative studies across a variety of developmental and evolutionary conditions. We therefore turned to the use of bulk RNA-sequencing (RNA-seq) analysis of purified TECs as a more versatile analytical approach. To identify relative shifts among canonical and mimetic TEC signatures, we performed competitive enrichment analysis of specific cell signatures against relevant control conditions[25] (Fig. 1a,b and Supplementary Fig. 1). At E15.5 and P0, early progenitor and cTEC signatures were highly enriched compared with P28, whereas mTEC and most mimetic signatures were depleted (Fig. 1a,b), mirroring the scRNA-seq data (Extended Data Fig. 1c). By contrast, signature gene sets of ciliated, goblet, ionocyte and muscle cells showed less

variation over developmental time (Fig. 1a,b and Supplementary Fig. 2). These observations indicated that individual groups of mimetic cells may develop in an asynchronous fashion, possibly related to the successive activities of the previously identified early and postnatal TEC progenitors[20].

To explore this possibility further, we examined the expression levels of the *Foxn1* gene, which encodes a key transcription factor that is required for the differentiation of canonical TECs[26-28], in the major TEC populations and mimetic cells. At P28, the cTEC population exhibits the highest proportion of *Foxn1*-expressing cells (Extended Data Fig. 3a). When compared with Aire-stage cells (Fig. 1c), fewer mimetic cells express *Foxn1*, although some differences exist among the various mimetic cell types (Extended Data Fig. 3b). Using a *Foxn1*-directed CRISPR–Cas9 barcoding system[20], we found that both canonical TEC populations and mimetic cells were marked by the presence of barcodes (Extended Data Fig. 3c) in similar proportions (Fig. 1d); as only few muscle cells could be detected in this experiment, no conclusion can be reached for this cell type. When analysed at the level of individual informative barcodes, it becomes apparent that mimetic cells as a group share their barcodes more often (but not exclusively) with postnatal progenitors than with early progenitors (Extended Data Fig. 3c). This difference can be explained by the small number of mimetic cells such as muscle cells that belong to the developmentally early group, and the large expansion of the postnatal progenitor population and its mimetic descendants, such as enterohepatic and skin cells (Extended Data Fig. 1c). The spatial distribution of mimetic cells as analysed by multicolour RNA ISH suggests a close affinity of *Aire*-expressing cells to the *Igfbp5*-expressing[20,29] postnatal progenitor population, as expected; the same is true for *Foxi1*-expressing ionocytes, whereas *Foxj1*-expressing ciliated cells are not found in close vicinity to the presumptive postnatal progenitor population (Extended Data Fig. 4). Likewise, reconstruction of the developmental trajectory of mimetic cells with CellRank[29,30] identified some mimetic cell types, such as ciliated cells (Extended Data Fig. 5 and Supplementary Fig. 3) and muscle cells (Supplementary Fig. 3), as early branching mimetic cell types. We conclude that mimetic cells and canonical TECs initially pass through a developmental stage marked by *Foxn1* gene expression but subsequently follow different developmental trajectories.

## Genetic determinants of mimetic cells

We sought additional clues to support the hypothesis of a developmentally orchestrated appearance of mimetic cells in the thymic microenvironment by analysis of a number of genetic models. To this end, we first explored whether the presence and proportion of mimetic cells is affected by mouse strain and size of the thymus (https://phenome.jax.org/measureset/10415) (Supplementary Fig. 4). Compared with strains such as C57BL/6 or CBA, the thymus of PWK mice supports approximately half of the number of thymocytes (Fig. 2a), and about 5% the number of TECs (Fig. 2b). The thymi of F₁ hybrids of CBA and PWK mice exhibit intermediate values for these parameters[31] (Extended Data Fig. 6). Phenotyping of reciprocal backcrosses revealed clear separations in the distributions for thymocytes and TEC numbers (Extended Data Fig. 6), suggesting that thymopoietic activity is determined by only few large-effect modifiers. Comparative analysis of TEC transcriptomes indicated only subtle differences; notably, whereas an enrichment of the cTEC signature was observed, no significant differences among mimetic cell types were detected when comparing PWK mice to C57BL/6 mice (Fig. 2c, Extended Data Fig. 7a and Supplementary Fig. 5a). Thus, despite the size differences of the organ, the representation of mimetic cells in the thymic microenvironment appears to be unchanged.

Given the known role of *Foxn1* for the TEC differentiation and maturation processes, we next examined the phenotype of *Foxn1*⁺/⁻ heterozygous mice, as *Foxn1* haploinsufficiency is known to be associated with

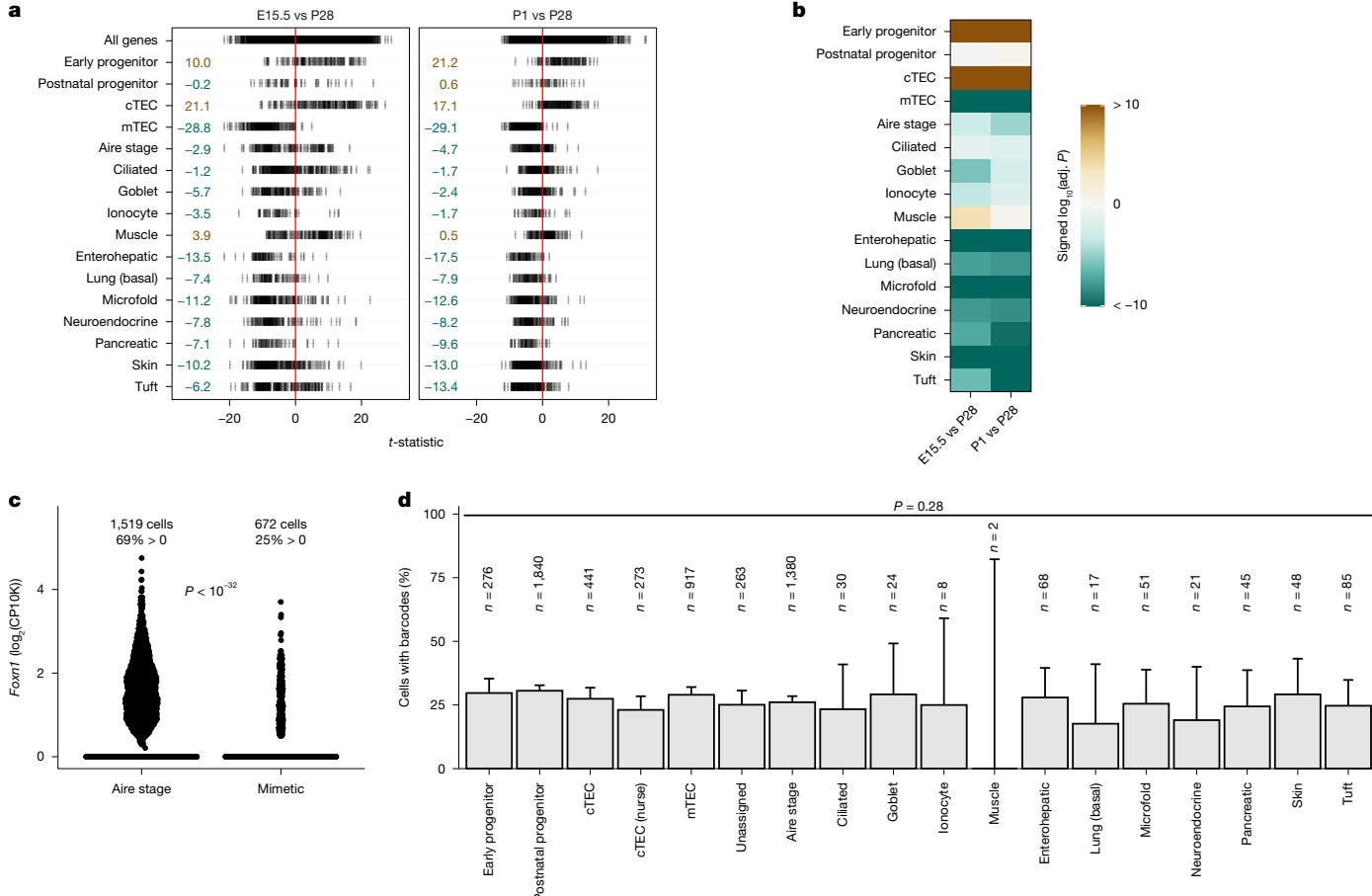

**Fig. 1 | Development of mouse thymic mimetic cells.** Enrichment of canonical TEC and mimetic signatures in bulk RNA-seq data of purified TECs from E15.5 ($n$ = 4) and P1 ($n$ = 2) time points compared with the P28 ($n$ = 3) time point. **a**, Visualization of changes in gene expression between conditions. Each line represents a gene of the indicated signature, and its position on the $x$ axis shows the value of the $t$-statistic derived from differential expression analysis. Numeric values listed in the left column represent $\log_{10}$(adjusted $P$) from enrichment analysis with camera[25]. Log-transformed $P$ values of upregulated sets were multiplied by −1 so that positive values indicate upwards directionality and negative values indicate downwards directionality. **b**, Log-transformed and signed $P$ values from camera (two-sided, Benjamini–Hochberg adjusted)

as shown in **a**, represented as a heat map. Values beyond the limits of the colour scale were rounded to the nearest limit. Adj., adjusted. **c**, Overall *Foxn1* expression levels ($\log_2$ of counts per 10,000 reads (CP10K)) in Aire-stage cells and mimetic cells from scRNA-seq data ($n$ = 4 mice; age, P28). The proportion of cells with detectable *Foxn1* is indicated; $P = 1.2 \times 10^{-33}$ (two-sided binomial test, Benjamini–Hochberg adjusted). **d**, Proportion of cells with detectable CRISPR–Cas9-induced barcodes at the *Hprt* locus for each signature. $n$ indicates total number of cells, pooled from three mice at P28. Data are mean ± 95% confidence interval of proportions (Wilson/Brown method[51]); $P$ values are derived via likelihood ratio test between logistic regressions with or without 'signature' as a predictor.

a smaller thymus[32]. We observed a relative increase of early progenitor and cTEC signatures, a less pronounced enrichment of the postnatal progenitor signature, and a depletion of the pan-mTEC signature, compatible with a partial block of TEC differentiation (Fig. 2d, Extended Data Fig. 7b and Supplementary Fig. 5b). Among the mimetic cell signatures, a reduction was observed for skin (Fig. 2d), compatible with the known extrathymic function of *Foxn1* in the differentiation of keratinocytes[33]. However, other mimetic signatures and the Aire-stage mTECs were either unchanged or only marginally affected (Fig. 2d). We conclude that only minor quantitative—but no qualitative—changes occur in the mimetic cell pool in *Foxn1*-heterozygous mice.

To explore the developmental origin of the mimetic cell compartment further, we used additional models that were suitable for genetic interference with the maturation of the thymic microenvironment. First, we overexpressed *Bmp4* under the control of the *Foxn1* promoter (tg*Bmp4;Foxn1*$^{+/+}$) to induce an immature thymic microenvironment[34,35], and analysed the relative shifts of mimetic cell signatures in the transgenic thymus at P28. Compared with non-transgenic mice, the signature of the early progenitor population was increased, as expected (Fig. 2e, Extended Data Fig. 7c and Supplementary Fig. 5c). However, changes in

the gene signatures of the different mimetic cell populations were again not uniform; ciliated, goblet and muscle cell signatures were enriched, whereas others remained largely unchanged (Fig. 2e), reminiscent of the relative increase in the former group of mimetic signatures at earlier time points in non-transgenic mice (Fig. 1b). Second, using *Ascl1*$^{-/-}$; *Foxn1*$^{+/+}$ mice, we examined the effect of TEC-specific inactivation of *Ascl1*, which encodes a neuronal tissue-specific transcription factor[36] that is also highly expressed in cells of the postnatal progenitor population (Extended Data Fig. 7d). In keeping with the expression profile of *Ascl1* (Extended Data Fig. 7d), we observed a shift among the canonical TEC signatures; the mTEC signature was decreased, whereas the cTEC and both progenitor cell signatures were correspondingly increased. Under these conditions, the greatest reductions among mimetics were observed for neuroendocrine and tuft cell signatures as noted previously[6]. This finding is indicative of an affinity of these cell types to the neuronal lineage (Fig. 2e, Extended Data Fig. 7c and Supplementary Fig. 5c). Third, we examined the effect of *Fgf7* overexpression (via tg*Fgf7;Foxn1*$^{+/+}$ mice) in the thymic microenvironment[20], which in accordance with the expression profile of the cognate *Fgfr2* receptor gene (Extended Data Fig. 7d), led to an enrichment of progenitor

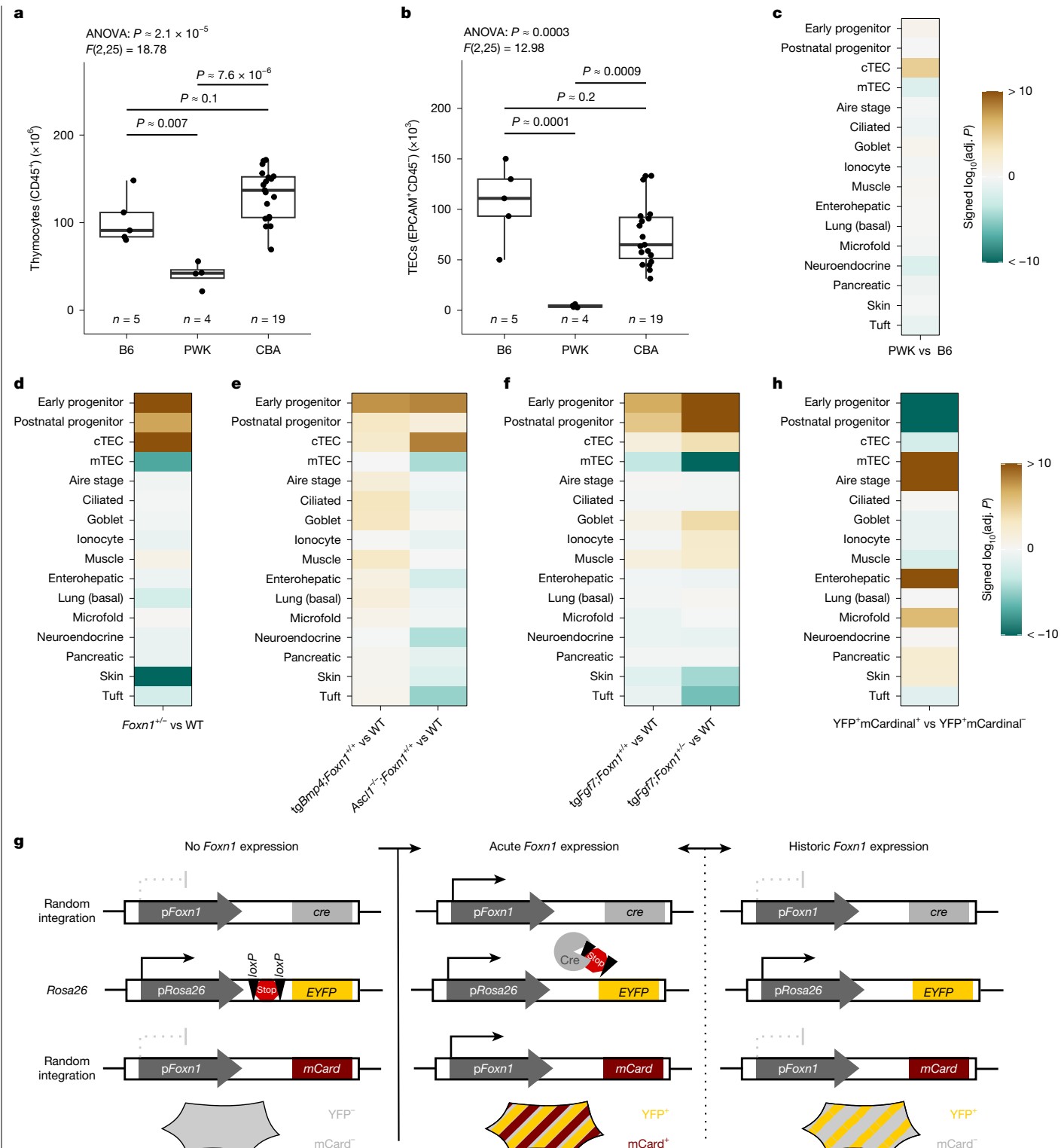

**Fig. 2 | Malleability of mimetic cell populations.** Thymic cellularity of C57BL/6 (B6), PWK and CBA strains at P28. **a**, Absolute numbers of CD45⁺ thymocytes. **b**, Absolute numbers of EPCAM⁺CD45⁻ TECs. **a**,**b**, $n$ is indicated in panels; each data point represents one thymus explant from one animal. $P$ values between groups were derived from pairwise two-sided $t$-tests with Bonferroni correction after significant ($P < 0.05$) ANOVA. Boxes encapsulate the first to third quartile, the line indicates the median and whiskers extend to the furthest point with a distance of up to 1.5 times the interquartile range from the boxes. **c**–**f**,**h**, Signature enrichment of canonical TEC and mimetic signatures in bulk RNA-seq data of purified TECs for PWK versus C57BL/6 (**c**), *Foxn1*⁺/⁻ versus wild type (WT) (**d**), tg*Bmp4;Foxn1*⁺/⁺ and *Ascl1*⁻/⁻;*Foxn1*⁺/⁺ versus wild type (**e**),

tg*Fgf7;Foxn1*⁺/⁺ and tg*Fgf7;Foxn1*⁺/⁻ versus wild type (**f**) and YFP⁺mCardinal⁺ versus YFP⁺mCardinal⁻ (**h**). Wild-type PWK, $n = 4$; wild-type C57BL/6, $n = 3$; *Foxn1*⁺/⁻, $n = 4$; tg*Bmp4;Foxn1*⁺/⁺, $n = 3$; *Ascl1*⁻/⁻, $n = 3$; tg*Fgf7;Foxn1*⁺/⁺, $n = 3$; tg*Fgf7;Foxn1*⁻/⁻, $n = 3$; YFP⁺mCardinal⁺, $n = 2$; YFP⁺mCardinal⁻, $n = 2$. **c**–**h**, Values beyond the limits of the colour scale were rounded to the nearest limit. Adj., adjusted. **g**, Schematic illustrating the principle of the YFP/mCardinal dual reporter system. Activity of the *Foxn1* promoter at any time point during development results in Cre recombinase expression and permanent activation of YFP expression. Acute activity of the *Foxn1* promoter is assessed by the mCardinal (mCard) reporter.

signatures and a decrease in the mTEC signature[20], whereas other signatures were not significantly affected (Fig. 2f, Extended Data Fig. 7e and Supplementary Fig. 5d). Combining the effects of *Fgf7* overexpression and *Foxn1* haploinsufficiency (via tg*Fgf7;Foxn1*[+/−] mice) exacerbated the maturation block of TECs, accompanied by a depletion of skin and tuft cell signatures (Fig. 2f, Extended Data Fig. 7e and Supplementary Fig. 5d). Collectively, these results support the notion of unexpected developmental heterogeneity among mimetic cell types and their malleability through extrinsic signals.

Although the precursors of mimetic cells pass through a stage of *Foxn1* gene expression (Fig. 1d), their development and/or maintenance may not depend on the activity of the FOXN1 transcription factor itself. To address this possibility, we turned to a triple-transgenic mouse model[20], which enabled us to distinguish TECs that were actively transcribing the *Foxn1* gene from those that ceased expression of this gene (Fig. 2g). Activation of the *Foxn1* gene during any time of TEC development indelibly marks such cells with YFP fluorescence. This is achieved by the combination of two transgenes; the *Foxn1* promoter drives the expression of the Cre recombinase, which results in the excision of a stop cassette in the *Rosa26* locus to allow YFP gene expression. The additional presence of a *Foxn1*-mCardinal transgene marks cells with red fluorescence as a sign of acute *Foxn1* promoter activity (Fig. 2g). RNA-seq analysis of YFP[+]mCardinal[+] (acutely *Foxn1*-expressing) TECs and YFP[+]mCardinal[−] (acutely *Foxn1*-negative, but with a history of *Foxn1* expression) TECs indicated a preponderance of the mTEC gene signature in the former; correspondingly, gene signatures characteristic for cTECs and the two progenitor populations were underrepresented (Fig. 2h, Extended Data Fig. 7f and Supplementary Fig. 6a). Likewise, Aire-stage, enterohepatic, microfold, pancreatic and skin gene signatures were enriched in YFP[+]mCardinal[+] TECs, whereas ciliated, goblet, ionocyte, lung (basal), neuroendocrine and tuft signatures were not grossly changed, or slightly reduced (muscle cell signature) (Fig. 2h and Extended Data Fig. 7f), hinting at a cell-specific effect of FOXN1 transcription factor levels for mimetic cell development.

## *Foxn1* gene and mimetic cell development

Previously, we showed that two distinct progenitor populations contribute to the thymic microenvironment[20], a cTEC-biased early progenitor, and an mTEC-biased postnatal progenitor. Analysis of the cTEC/mTEC ratio via flow cytometry, using Ly51 (also known as CD249) and *Ulex europaeus* agglutinin-1 (UEA1) as markers for cTECs and mTECs, respectively, indicated a prominent nadir at around 3 weeks of age (P21) (Fig. 3a). We reasoned that this inflection point might mark the transition period during which the TEC compartment becomes less dependent on the early progenitor and more reliant on its postnatal counterpart.

When the endogenous *Foxn1* gene is replaced in mice by its more ancient paralogue *Foxn4*, the demarcation of cortex and medulla is impaired[37,38]. Indeed, domain-swap experiments demonstrated that sequences in the second coding exon of *Foxn1* are particularly important to establish the maturation of the epithelial compartment[38], and thus may be required for the developmental transition from early to postnatal TEC progenitor activities. A multiple sequence alignment of FOXN1 proteins from representative species of jawed vertebrates indicated that the sequences encoded by the 3′ part of this exon are common to FOXN1 proteins of all jawed vertebrates (Fig. 3b). Accordingly, we generated a mouse strain (Δ3ex2) that expresses a modified FOXN1 protein lacking this domain (Fig. 3c). In this strain, the embryonic and perinatal development of the thymus proceeded normally; however, thymopoietic activity collapsed in the third postnatal week, coincident with the nadir in Ly51/UEA1 ratios. Principal component analysis of transcriptomes of purified TECs collected at this time point separated the mice into two groups (Fig. 3d), mirrored in altered thymic cellularities (Fig. 3e and Extended Data Fig. 8a) that are suggestive of inactive

(collapsing) and active (recovering) microenvironments. Enrichment analysis of cell-type-specific signatures indicated that during the phase of diminished thymopoietic activity (Extended Data Fig. 8a and Supplementary Fig. 6b), the epithelial compartment exhibits signs of a bias towards an immature stage, with a prominent enrichment of the early progenitor in conjunction with depletion of mTEC and Aire-stage signatures (Fig. 3f). Of note, with respect to mimetic cell signatures, we found that the signatures of enterohepatic, microfold, pancreatic and skin cells were significantly reduced (Fig. 3f and Extended Data Fig. 8b), again highlighting the association of ongoing mTEC maturation and their development. The mimetic cell composition of thymopoietically active thymi (that is, those in the recovery phase) was essentially indistinguishable from the wild type, accompanied by a slight increase in early progenitor and cTEC signatures (Fig. 3f, Extended Data Fig. 8b and Supplementary Fig. 6c), strongly suggesting that only some mimetic cell types (for instance, those with an enterohepatic signature) arise from the postnatal progenitor population.

Next, we directly examined the possibility that the degrees of dependence on FOXN1 transcription factor activity may substantiate the classification of mimetic cell types. To this end, we isolated TECs from *Foxn1*-deficient mice. As expected, mTEC and Aire-stage signatures were clearly reduced, alongside enterohepatic, microfold, pancreatic, skin and tuft cell signatures (Fig. 4a, Extended Data Fig. 9a and Supplementary Fig. 7). By contrast, expression levels of signature genes characteristic of other mimetic cells, such as ciliated, goblet and muscle cells, did not differ from those of *Foxn1*-wild-types, whereas those of ionocytes were increased (Fig. 4a and Extended Data Fig. 9a). The results of RNA ISH experiments confirmed the presence of cellular heterogeneity in the *Foxn1*-deficient epithelium (Fig. 4b and Extended Data Fig. 9b). For instance, the expression pattern of *Foxi1*, encoding a key transcription factor for ionocytes[39], overlapped at least partially with that of *Foxn1* (Fig. 4b). By contrast, cells expressing *Foxj1*, encoding the key transcription factor of ciliated cells[40], were found in areas devoid of *Foxn1* gene activity, suggesting that the FOXN1 transcription factor is not required for their development in the thymic anlage[41]. To directly assess whether the ciliated cells in the *Foxn1*-deficient thymic rudiment match their counterpart in the *Foxn1*-sufficient thymic microenvironment, we determined their transcriptional signatures by single-nucleus RNA-seq (snRNA-seq) (Extended Data Fig. 9c). Cells compatible with a ciliated signature formed a singular cluster comprising cells from both genotypes, indicative of highly similar overall transcriptomic landscapes (Fig. 4c). This indicates that the differentiation of ciliated cells in the thymic microenvironment is independent of FOXN1 transcription factor activity. Owing to the distorted differentiation of *Foxn1*-deficient epithelium (Extended Data Fig. 9c), other TEC subsets, such as ionocytes, are more difficult to reliably compare between genotypes (Fig. 4d and Extended Data Fig. 9d,f). Unlike *Foxj1* and *Foxi1*, expression of *Hnf4a*, a key transcription factor of the enterohepatic lineage, was not detectable in the *Foxn1*-deficient epithelium (Fig. 4b), indicating that development of enterohepatic cells[5] depends, at least transiently, on FOXN1 transcription factor activity. Thus, whereas most mimetic cell types require *Foxn1* activity for their development, some cell types, such as ciliated cells, can develop in the absence of FOXN1.

## Evolutionary history of mimetic cells

To explore the evolutionary history of mimetic cells, we examined the thymus of the cartilaginous fish *Scyliorhinus canicula* as a representative of the most basal jawed vertebrate group. We examined the presence and location of cells expressing characteristic mimetic genes (Extended Data Fig. 10a and Supplementary Fig. 8) in the thymus (Fig. 5a–c and Extended Data Fig. 10b) by RNA ISH. Similar to the situation in mice[2] (Extended Data Fig. 2) and zebrafish[7], putative mimetic cells, such as those marked by the expression of *TTR* (a liver-specific gene[42]), *FOXI1*, *FOXJ1* and *POU2F3* (a marker of tuft cells) were found in

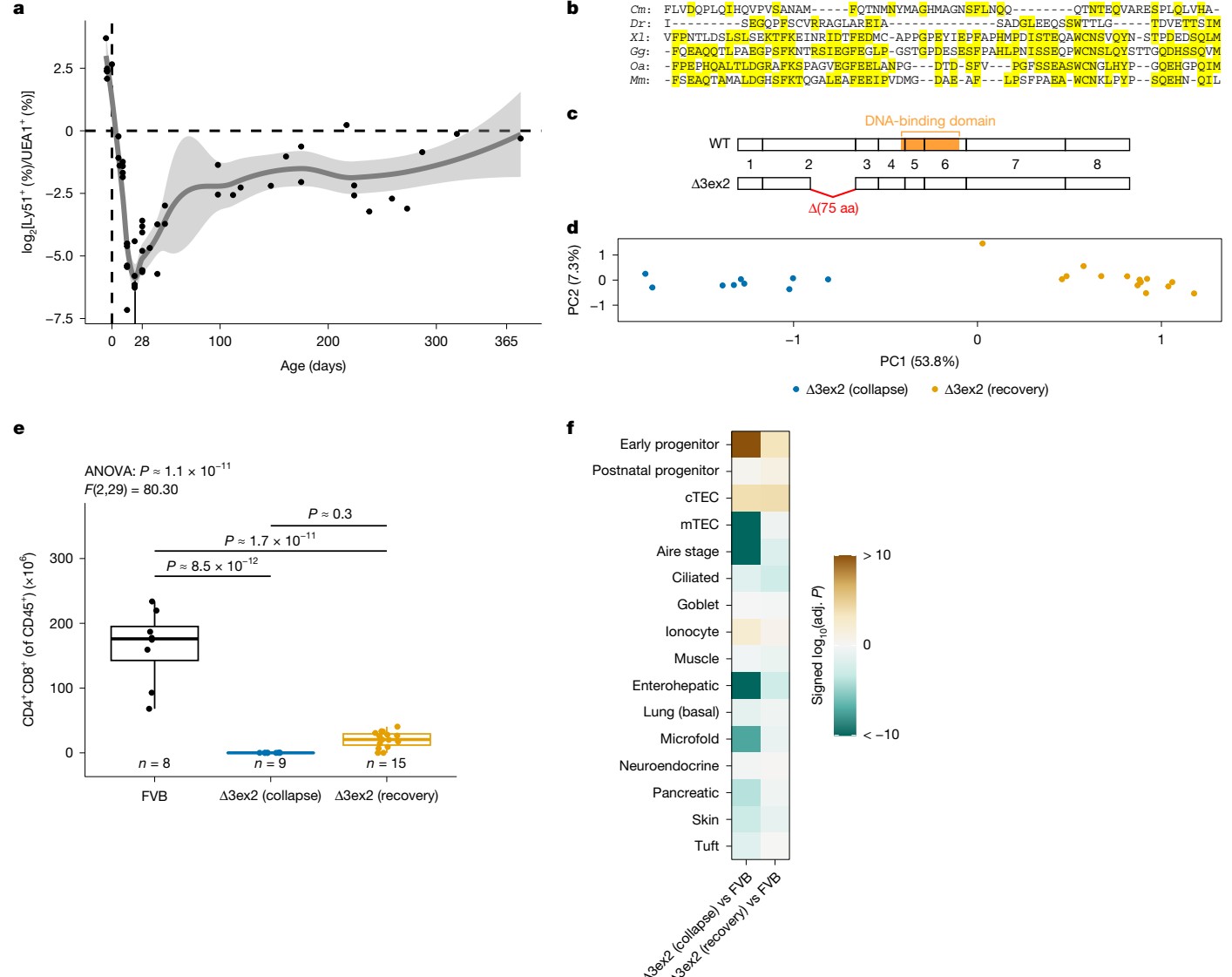

**Fig. 3 | FOXN1 influences the development of mimetic cells. a**, Ratio of Ly51[+] (cTEC) and UEA1[+] (mTEC) EPCAM[+]CD45[−] TECs cells over developmental time. Each point shows the ratio from one thymus. The trend line was determined via LOESS with tenfold cross-validation. The minimum occurs at P21. Error bands show 95% confidence interval around the predicted trend line. **b**, Multiple sequence alignment of the C-terminal end of FOXN1 protein sequences encoded by exon 2 of the gene in various jawed vertebrate species; identical amino acid residues are shaded. *Cm*, *Callorhinchus milii*; *Dr*, *Danio rerio*; *Xl*, *Xenopus laevis*; *Gg*, *Gallus gallus*; *Oa*, *Ornithorhynchus anatinus*; *Mm*, *Mus musculus*. Sequences obtained from ref. 38. **c**, Schematic illustrating the coding content of *Foxn1* exons and the deleted 3′ region of *Foxn1* coding exon 2 in the Δ3ex2 mutant; three exons contribute to the DNA-binding domain as indicated. **d**, Principal component analysis of bulk RNA-seq samples from purified TECs of Δ3ex2 mutants, collected from mice around P21. **e**, Numbers of CD4/CD8-double positive thymocytes in thymi of wild-type FVB mice and Δ3ex2 mutants during collapse and recovery phases. *n* is indicated; each data point is one thymus explant from one animal. *P* values between groups were derived from pairwise two-sided *t*-tests with Bonferroni correction after significant (*P* < 0.05) ANOVA result. Boxes encapsulate the first to third quartile, the line indicates the median and whiskers extend to the furthest point with a distance of up to 1.5 times the interquartile range from the boxes. **f**, Signature enrichment analysis in bulk RNA-seq data of purified TECs in Δ3ex2 mice (FVB (wild type), *n* = 6; collapse, *n* = 9; recovery, *n* = 15). *P* values from camera (two-sided, Benjamini–Hochberg adjusted) were log-transformed and multiplied by −1 for upregulated sets, so that positive values indicate upwards directionality and negative values indicate downwards directionality. Values beyond the limits of the colour scale were rounded to the nearest limit. Adj., adjusted.

the medullary region (Fig. 5d,e and Extended Data Fig. 10c). *FOXN1* and its paralogue *FOXN4* were expressed in the thymic microenvironment of cartilaginous[43] (Fig. 5c and Extended Data Fig. 10c) and bony fishes[37].

We previously hypothesized that a primordial thymopoietic activity of vertebrates was driven by *Foxn4*, the ancestor of *Foxn1*[37,38]. We suggest that the emergence of new cell types and organs, such as the liver, in the ancestor common to all vertebrates[12], was accompanied by the transition from an invertebrate form of *Foxn4* (via a primordial vertebrate version of *Foxn4*) to its vertebrate-specific paralogue

*Foxn1* to achieve the necessary adaptation of central tolerance mechanisms. Because genetic manipulation of cartilaginous fishes is not possible, we examined the individual thymopoietic capacities of the shark *Foxn1* and *Foxn4* genes using a *Foxn1*-replacement model. In this system, the mouse *Foxn1* gene is replaced by other evolutionarily distinct members of the *Foxn1/4* gene family[38]. The TEC compartments driven by the shark *Foxn1* and *Foxn4* gene of a cartilaginous fish (*C. milii*), the most basal vertebrate group, are not identical (Fig. 5f and Extended Data Fig. 10d), although in *Foxn1;Foxn4* double-transgenic

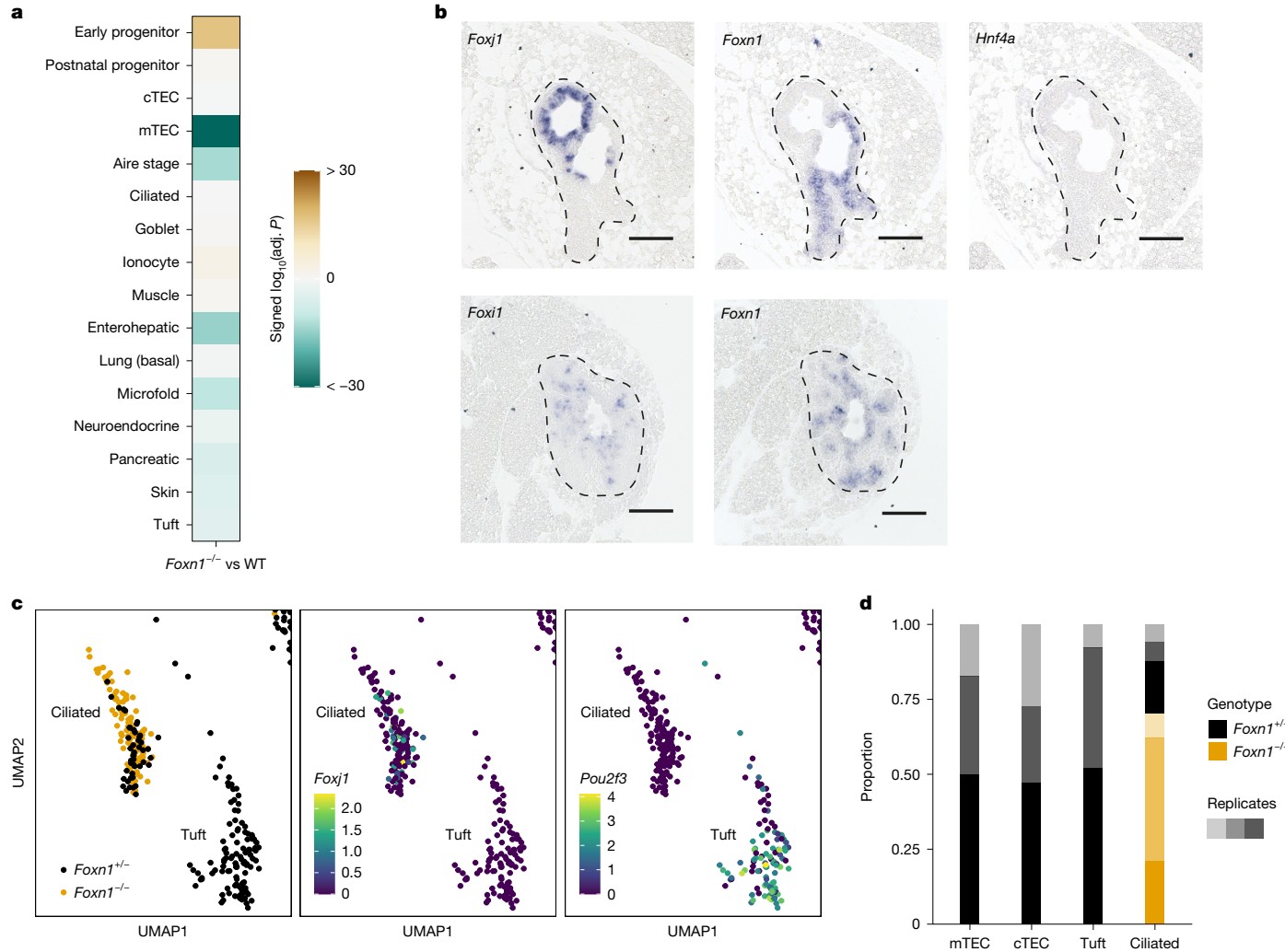

**Fig. 4 | Requirement of _Foxn1_ for mimetic cell development. a**, Signature enrichment analysis in bulk RNA-seq data of purified TECs of _Foxn1_[−/−] mutants compared with wild-type controls (wild type, _n_ = 3; _Foxn1_[−/−], _n_ = 4). _P_ values from camera (two-sided, Benjamini–Hochberg adjusted) were log-transformed and multiplied by −1 for upregulated sets, so that positive values indicate upwards directionality, and negative values indicate downwards directionality. Values beyond the limits of the colour scale were rounded to the nearest limit. Adj., adjusted. **b**, Micrographs of thymic rudiments in _Foxn1_[−/−] mice after RNA ISH with the indicated probes; top and bottom rows depict consecutive sections. Cells expressing the indicated genes are labelled in blue. Data representative of five mice. Scale bars, 0.1 mm. **c**, Characterization of the ciliated and tuft cell clusters (see Extended Data Fig. 9c). Each dot represents a cell, with the genotype of origin indicated by colour (left). Expression levels of _Foxj1_ (middle) and _Pou2f3_ (right) are provided as log[2]-normalized counts. **d**, Contributions of individual samples to the indicated clusters; genotypes are coloured and replicates are identified by different shades.

thymi (resembling the physiological situation in the shark thymus), the mimetic cell composition is more similar to the mouse wild type than is either of the two single-transgenic compartments (Fig. 5f and Extended Data Fig. 10d). The results of the reconstitution experiment thus suggest that FOXN4 and FOXN1 transcription factors in the shark thymic epithelium non-redundantly contribute to achieve coverage of peripheral cell types.

We directly tested the non-redundancy by examining the composition of the thymic microenvironment in teleosts lacking the _foxn1_ gene, the evolutionarily younger paralogue of _foxn4_[43]. The mimetic signatures in the mutant zebrafish thymus can be categorized into two groups (Fig. 5g); goblet, ionocyte, muscle, lung (basal), neuroendocrine and skin signatures were enriched compared with those representing enterohepatic, microfold, pancreatic, tuft and ciliated cells (Fig. 5g, Extended Data Fig. 11a and Supplementary Figs. 7 and 8). This phenotype is similar to the bias introduced by _Foxn4_ in the shark replacement model (Fig. 5f). In the teleost thymus, additional aspects of central tolerance formation are also dependent on the evolutionarily younger FOXN1 transcription factor. For instance, expression of _Psmb11_, which

encodes a thymus-specific component of the proteasome and is important for positive selection of CD8[+] thymocytes[44], is undetectable in the _Foxn4_-driven microenvironment (Extended Data Fig. 11b), in line with the requirement of FOXN1 for _Psmb11_ expression in the mouse thymus[45]. Moreover, despite the fact that the Aire-stage cell signature remains unchanged (Fig. 5g), the expression levels of _aire_ itself are nonetheless reduced in the _foxn1_-deficient microenvironment (Extended Data Fig. 11b). Collectively, these data suggest that the contribution of _foxn1_ to central tolerance formation with respect to supporting the presence of mimetic cells appears to be biased towards evolutionarily more recent tissue and cell types.

To substantiate this conclusion, we examined the thymopoietic activity of the _Foxn4_ gene of amphioxus (_Branchiostoma lanceolatum_). Amphioxus is a representative of cephalochordates, a distant relative of vertebrates without adaptive immunity. In the _Foxn1_-replacement model, the microenvironment driven by _B. lanceolatum_ Foxn4 supports T cell development only up to the CD4/CD8-double positive stage, and these mice exhibit autoimmune phenomena[38], compatible with the pre-adaptive function of this transcription factor. Gene

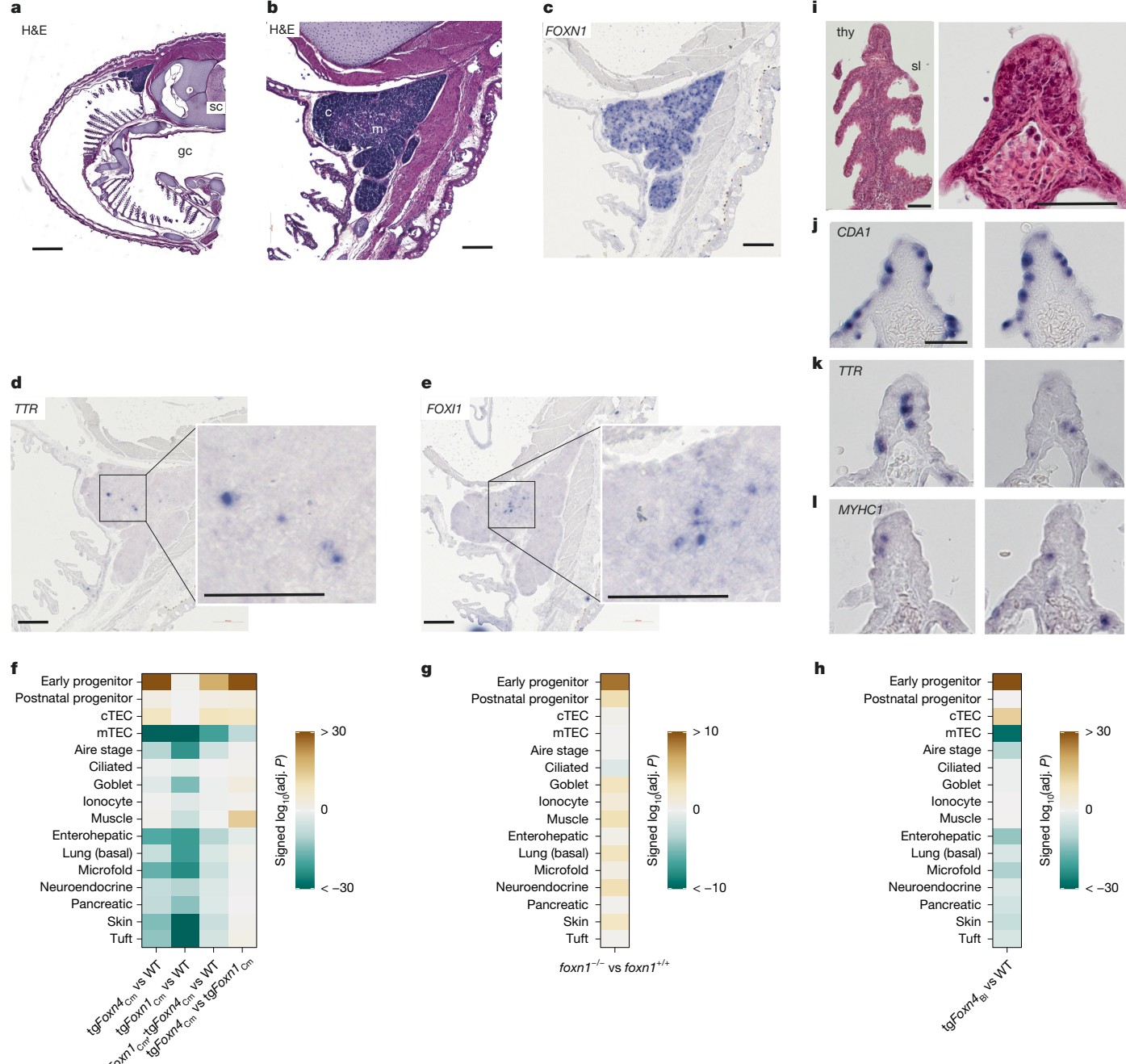

**Fig. 5 | Evolutionary trajectory of mimetic cells. a**, Macroscopic view of the gill basket of a juvenile *S. canicula* specimen; gc, gill chamber; sc, spinal chord; H&E, haematoxylin and eosin. **b**, Higher magnification of the thymus region in **a**, showing the cortical (c) and medullary (m) structures of the thymus. Images in **a**,**b** are representative of *n* = 3 animals. **c**, Micrographs of the shark thymus after RNA ISH with *FOXN1* (blue). **d**,**e**, Micrographs of the shark thymus after RNA ISH with *TTR* (**d**) and *FOXI1* (**e**). Images on the right show the medullary regions at higher magnification. **c**–**e**, Images are representative of *n* = 2 animals. **f**, Signature enrichment analysis of mouse *Foxn1*−/− mutants expressing ancient *Foxn1* and *Foxn4* genes under the control of the mouse *Foxn1* promoter compared against corresponding wild-type controls. tg*Foxn4*Cm, *Foxn4* gene from the cartilaginous fish *C. milii* (*n* = 4); tg*Foxn1*Cm, *Foxn1* gene from *C. milii* (*n* = 3); tg*Foxn1*Cm,tg*Foxn4*Cm (double-transgenic mice, *n* = 5); wild type, *n* = 3. The last

column represents a comparison of TECs from the two single-transgenic strains. **g**, Signature enrichment analysis in bulk RNA-seq of whole thymi from *foxn1*−/− zebrafish (*n* = 3) compared with *foxn1*+/+ wild types (*n* = 3). **h**, Signature enrichment analysis of mouse *Foxn1*−/− mutants expressing the *Foxn4* gene from the cephalochordate *B. lanceolatum* (*n* = 4); wild type, *n* = 3. **i**, Left, micrograph depicting a gill filament of *Lampetra planeri* (H&E staining). thy, thymoid; sl, secondary lamellae. Right, further magnified view of the thymoid, indicating the tissue heterogeneity (H&E staining); the blood vessel is filled with nucleated erythrocytes. Image representative of *n* = 20 animals. **j**–**l**, Micrographs of thymoids after RNA ISH with probes specific for *CDA1* (**j**), *TTR* (**k**) and *MYHC1* (**l**). Rows depict consecutive sections; images are representative of *n* = 3 animals. Scale bars: 0.1 mm (**a**); 0.2 mm (**b**–**e**); 0.1 mm (**i**–**l**).

signature analysis of reconstituted TECs revealed an enrichment of the early progenitor and cTEC signatures, and a paucity of mTEC and Aire-stage cells (Fig. 5h, Extended Data Fig. 11c and Supplementary

Fig. 7), indicative of blocked maturation of the epithelium. Notably, whereas gene signatures for ciliated, goblet, ionocyte and muscle cells were not significantly different from those of wild-type TECs, we noted a

distinct reduction of other mimetic cell signatures, which in the mouse environment likely causes incomplete representation of peripheral self antigens. Indeed, with the exception of the notable presence of the characteristic cTEC signature, the pattern in *B. lanceolatum Foxn4* transgenic mice (Fig. 5h) is essentially indistinguishable from that of *Foxn1*-deficient mice (Fig. 4a), in which T cell development is completely blocked and hence no autoimmunity is observed. We conclude that also in this evolutionary reconstruction, the group of ciliated, goblet, ionocyte and muscle cell behaves distinctly differently from the other mimetic cell types, revealing an evolutionarily plausible trend of stepwise addition of mimetic cell types. The development of certain mimetic cells, such as the enterohepatic group, are clearly dependent on the formation of the mTEC compartment, the full maturation of which is driven by FOXN1. This group of mimetic cells represents cell types that are vertebrate-specific innovations (no liver is detectable outside the vertebrate lineage[12]), whereas other mimetic cell types, such as goblet, muscle and others, are representatives of evolutionarily more ancient cell types and thus may have been added to the portfolio of mimetic cells at an earlier stage of vertebrate evolution.

Considering the present results and earlier work[2,7], peripheral antigen expression in the thymus appears to be a feature of all jawed vertebrates, ranging from sharks to humans. To explore the evolutionary origin of this central tolerance mechanism further, we turned our attention to lampreys, the best-studied representative of jawless vertebrates, the sister group of jawed vertebrates. To this end, we determined whether cells that express peripheral (tissue-restricted) antigens are present in the thymoid, the thymus equivalent of lampreys, which is situated at the tip of gill filaments and distal secondary lamellae. Although the anatomical structure of the thymoid in lampreys is less well studied than that of the thymus of jawed vertebrates, it is clearly a structured tissue (Fig. 5i). The expression of *CDA1*, which encodes a cytidine deaminase, identifies developing lamprey T-like cells in the distal (outer) area of the thymoid[46], a localization that is reminiscent of the thymic cortex of jawed vertebrates (Fig. 5j). By contrast, cells that express the *TTR* gene, which is most highly expressed in the lamprey liver[47], are predominantly situated in the inner region of the thymoid (Fig. 5k). We also detected cells that express the gene for myosin heavy chain (*Myhc1*; Fig. 5l). Thus, our results raise the intriguing possibility that intrathymic tolerogenic mechanisms, including self-antigen-displaying mTECs and/or thymic mimetic cells, are present in both jawless and jawed vertebrates.

## Conclusion

In mice, AIRE-mediated peripheral antigen expression by thymic epithelial cells[22], the activity of *Fezf2* in mTECs[23], and the presence of mimetic cells[2] in the thymic microenvironment non-redundantly contribute to the formation of a self-tolerant T cell repertoire. The *Aire* gene is found only in the genomes of jawed vertebrates[48] and is not present in jawless vertebrates; by contrast, *Fezf2* is a pan-vertebrate-specific gene (a paralogue of the evolutionarily ancient *Fezf1* gene[49]) found in both jawed and jawless vertebrates (Extended Data Fig. 11d). These findings indicate the possibility that *Fezf2*-driven peripheral self-antigen expression in the thymus might have emerged already in the ancestor common to all vertebrates, before AIRE-specific functions were added in jawed vertebrates. Thus, the tolerogenic capacity of the thymus might have evolved in successive steps, mirrored in the presence or absence of tolerogenic factors and the distinct developmental sequence of mimetic cell types.

In mice, FEZF2 and AIRE regulate only partially overlapping aspects of peripheral antigen expression[4,23]. Both tolerogenic factors—FEZF2 and AIRE—are required for the differentiation of some mimetic cells[2,4]. FEZF2 appears to be particularly relevant for the differentiation of the enterohepatic lineage of mimetic TECs[4]; notably, in mTECs of *Fezf2*-deficient mice, *Ttr*, which encodes transthyretin, is the most highly downregulated gene[23]. The presence of *TTR*-expressing cells

in the thymoid is compatible with the control of intrathymic *TTR* gene expression by FEZF2 rather than AIRE (which is absent from the lamprey genome). At present, the mechanistic underpinnings of antigen receptor repertoire development in the lamprey T cell lineage[50] are largely unexplored, complicating attempts at establishing the functional importance of peripheral antigen expression in the lamprey thymoid. Further work is required to establish the frequency and distribution of cells expressing different peripheral antigens in the gill filaments and to clarify the mechanism by which developing T cells gain access to these antigens. Nonetheless, our results reveal an unexpected similarity in tolerogenic traits between the two sister groups of vertebrates. Despite species-specific variations[7], the exposure of developing thymocytes to peripheral tissue antigens thus emerges as a general component of vertebrate adaptive immune systems.

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

## Methods

### Animals

C57BL/6 mice were maintained at the Max Planck Institute of Immunobiology and Epigenetics. *Foxn1*[−/−] (ref. 27), *Foxn1:cre*[52], *Rosa26-LSL-EYFP*[53], *Foxn1:mCardinal*[54], *Ascl1*[fl/fl] (ref. 55), *Foxn1:*Bl*Foxn4, Foxn1:*Cm*Foxn4, Foxn1:*Cm*Foxn1*[38] and *Foxn1:Fgf7*[20] transgenic mice have been described previously. The *Foxn1:Bmp4* transgene was created by T. Schlake and B.K. by inserting a cDNA fragment corresponding to nucleotides 497–1729 in GenBank accession number NM_007554.3 as a NotI fragment into pAHB14[34]. The Δ3ex2 *Foxn1* deletion mutant (internal designation Chi6) transgene was generated by deletion of nucleotides 504–728 of mouse *Foxn1* cDNA (Genbank accession number NM_008238.2) and insertion as a NotI fragment into pAHB14[34]. To generate transgenic mice, constructs were linearized and injected into FVB pronuclei according to standard protocols. The Foxn1[Δ3ex2] mice were bred to a *Foxn1*-deficient background. Genotyping information is summarized in Supplementary Table 2. Mice carrying the original *nu* mutation (CByJ.Cg-Foxn1nu/J) were purchased from Charles River and used for snRNA-seq experiments. Mice were analysed at the age of 4–6 weeks, unless otherwise stated.

The zebrafish line carrying an internal deletion of the *foxn1* gene has been described[56]. Adult zebrafish (3 months of age) were used for experiments.

Mice and zebrafish were kept in the animal facility of the Max Planck Institute of Immunobiology and Epigenetics under specific pathogen-free conditions (mice: 14 h light, 10 h dark; temperature 22 ± 2 °C; relative humidity 55 ± 10%; zebrafish: 13 h light, 11 h dark, water temperature 28 °C). All animal experiments were performed in accordance with the relevant guidelines and regulations, approved by the review committee of the Max Planck Institute of Immunobiology and Epigenetics and the Regierungspräsidium Freiburg, Germany (mice: licenses 35–9185.81/G-12/85; 35–9185.81/G-16/67; zebrafish: license 35–9185.81/G-14/41). All strains are made available from the corresponding author upon request, subject to standard material transfer agreements.

Ammocoete larvae of *L. planeri* (body length, 8–10 cm) were caught by a licensed fisherman from the wild in the Freiburg region (Riedgraben, March-Neuershausen) under permission by the local governmental authority (Landratsamt Breisgau-Hochschwarzwald, license 420.1.13-2024-034414). Juvenile cat shark (*S. canicula*) specimens were kindly supplied by Markéta Kauka (Max Planck Institute for Evolutionary Biology, Plön, Germany); juvenile bamboo sharks (*Chiloscyllium punctatum*) were purchased from a local pet shop. Upon arrival at the laboratory, lampreys and sharks were euthanized using 0.02% tricaine methanesulfonate. For RNA isolation, tissues were removed under a dissection microscope and dissolved in TRI reagent (T9424, Sigma-Aldrich); for histological studies, dissected tissues were fixed in 4% neutral formalin.

Sample sizes were based on our experience and accepted practice in the respective fields, balancing statistical robustness, resource availability and animal welfare. No statistical methods were used to predetermine sample size. Provided the transgenic status and age matched the experimental requirements, mice were randomly assigned to experimental groups, irrespective of sex. Blinding was not possible because the thymus phenotype—the transgenic status of the respective mouse—is evident from flow cytometry, imaging analysis or genotyping information.

### RNA in situ hybridization

Tissues were fixed with modified Davidson's fluid and dehydrated with Li-ethanol and ethanol in a stepwise manner and finally embedded in paraffin; RNA ISH on paraffin sections was performed using DIG-labelled probes[57]. Double ISH was carried out as follows. DIG- and fluorescein-labelled RNA antisense probes were simultaneously hybridized to RNA in tissue sections. The DIG-labelled probe was detected first, either with an alkaline-phosphatase-conjugated anti-DIG antibody (1:2,000 dilution in maleic acid buffer (MAB); 100 mM maleic acid, pH 7.5, 150 mM NaCl, 2 mM Levamisol, 1% blocking reagent (Roche), 0.1% Tween-20) for chromogenic detection, or with an anti-DIG-POD antibody (1:300 dilution in MAB) for fluorescent detection. The presence of the DIG-hapten was revealed by staining with BM Purple (Roche) or by Cy3 fluorescence, using the Tyramide Signal Amplification Plus system (AkoyaBioscience). The sections were washed several times in PBS; the fluorescein-labelled probe was detected by a peroxidase-conjugated anti-Fluorescein-POD (1:300 dilution in MAB) and revealed by Cy5 fluorescence.

Sequence coordinates in GenBank accession numbers for the probes were as follows:
- *M. musculus*: *Foxn1*, nucleotides 2181–3584 in XM_006532266.3; *Foxj1*, nucleotides 1406–1917 in NM_008240.3; *Hnf4a*, nucleotides 1469–1920 in NM_008261.3; *Foxi1*, nucleotides 1067–1576 in NM_023907.4; *Pax9*, nucleotides 1260–1818 in NM_011041.3.
- *S. canicula*: *RAG1*, nucleotides 3400–3900 in XM_038808256; *FOXN1*, nucleotides 101–600 in XM_038813127; *FOXN4*, nucleotides 2001–2500 in XM_038815571; *CD3E*, nucleotides 1–594 in KY434199; *FOXJ1*, nucleotides 1501–2000 in XM_038820881; *FOXI1*, nucleotides 1231–1730 in XM_038822317; *MYOG*, nucleotides 620–1120 in XM_038820992; *POU2F3*, nucleotides 1401–1900 in XM_038779605; *TTR*, nucleotides 180–680 in XM_038808583.
- *L. planeri*: *CDA1*, nucleotides 8–496 in MG495252; *TTR*, nucleotides 148–593 in XM_061573254; *MyHC1*, nucleotides 5013–5609 in AB126173.

### Image analysis

Images were acquired on Zeiss microscopes (Axioplan 2) equipped with an Mrc 5 camera; in some figure panels, Cy5 signals were converted to false (yellow) colour for better visualization.

### Flow cytometry and cell sorting

Single-cell suspensions of TECs for preparative flow cytometry were obtained as described[31,58]. Thymic epithelial cells have the surface phenotype EPCAM[+]CD45[−]; thus, cell surface staining was performed using anti-EPCAM (G8.8), conjugated with APC (1:1,000, BioLegend) or anti-EPCAM (G8.8), conjugated with biotin (1:1,000, BioLegend), in combination with streptavidin, conjugated with eFluor 450 (1:1,000, eBioscience), and anti-CD45 (30-F11), conjugated with PE Cy7 (1:2,000, BioLegend) at 4 °C in PBS supplemented with 0.5% BSA and 0.02% NaN$_3$. In order to differentiate cells with past and acute *Foxn1* expression, triple-transgenic *Foxn1:cre; Rosa26-LSL-eYFP; Foxn1:mCardinal* mice were used for cell sorting. Cells with past expression of *Foxn1* were sorted as EPCAM[+]YFP[+] mCardinal[−] cells, whereas cells with acute *Foxn1* expression were sorted as EPCAM[+]YFP[+] mCardinal[+] cells. EPCAM[+]CD45[−] cells (after negative enrichment using anti-CD45 magnetic-activated cell sorting (MACS) beads and anti–Ter-119 MACS beads, Miltenyi Biotec) were sorted directly into TRI reagent (T9424, Sigma-Aldrich). Cell sorting was carried out using the MoFlow instrument (Dako Cytomation-Beckman Coulter) controlled with the Summit (5.5) software. Analytical flow cytometry was performed for TECs as follows: anti-EPCAM (G8.8), conjugated with APC (1:1,000, BioLegend); anti-Ly51 (alias BP-1; 6C3), conjugated with PE (1:1,600, eBioscience); UEA1, conjugated with FITC (1:1,000, Vector Labs) or UEA1, conjugated with biotin (1:600, Vector Labs), in combination with streptavidin, conjugated with eFluor 450 (1:1,000, eBioscience). When analysis of haematopoietic fractions was desired, thymocyte suspensions were prepared in parallel by mechanical liberation, best achieved by gently pressing thymic lobes through 40 μm sieves. Cell surface staining (anti-CD45 (30-F11), conjugated with PE/Cy7 (1:2,000, BioLegend); anti-CD4 (GK1.5), conjugated with FITC (1:1,000, BioLegend); anti-CD8a (53-6.7), conjugated with

APC (1:800, eBioscience); anti-TCRβ (H57-597), conjugated with PE (1:400, eBioscience); anti-CD19 (eBio1D3), conjugated with PerCP/Cy5.5 (1:500, eBioscience) or PE/Cy7 (1:1,000, eBioscience); anti-B220 (alias CD45R; RA3-6B2), conjugated with biotin (1:200, eBioscience); anti-IgM (II/4.1), conjugated with PE (1:300, eBioscience), anti-CD93 (alias C1qRp; AA4.1), conjugated with APC (1:300, eBioscience); strepta-vidin conjugated with eFluor 450 or FITC (1:1,000, eBioscience)) was performed at 4 °C in PBS supplemented with 0.5% BSA and 0.02% NaN$_3$. Flow cytometry experiments were evaluated using FACSDiva (8.0.2) and FlowJo (9.3.1) software. The relevant gating strategies[20,37,38] are shown in Supplementary Fig. 4.

### Isolation of cell nuclei from thymus tissue
Thymus tissues were recovered under a dissection microscope; thymocytes were mechanically liberated by applying gentle pressure on the tissue on a 40-µm sieve and extensive washing to deplete as many haematopoietic cells as possible. For nude mice, thymic rudiments from two individuals were pooled to constitute one sample. The resulting tissue remnants were transferred into a 1.5 ml Eppendorf tube containing 150 µl of freshly prepared ice-cold lysis buffer (10 mM Tris-HCl pH 8, 10 mM NaCl, 3 mM MgCl$_2$, 0.1% non-denaturing detergent Igepal CA-630, 0.1% Tween-20, 1% BSA), supplemented with 1 U µl$^{-1}$ Protector RNase inhibitors (Roche, 3335402001). Tissues were homogenized using a RNAse-free disposable pestle for 1.5 ml tubes (Fisher Scientific, 12141364) with a circular motion until the majority of clumps were homogenized. Nuclei quality was assessed under a phase-contrast microscope. Subsequently, 500 µl of ice-cold wash buffer (10 mM Tris-HCl pH 8, 10 mM NaCl, 3 mM MgCl$_2$, 0.1% Tween-20, 1% BSA) with 0.2 U µl$^{-1}$ of Protector RNase inhibitors was added, mixed by inversion, and spun briefly to collect liquid from the cap of the tube. The nucleus suspension was then filtered through a 70 µm Flowmi Cell Strainer (H13680-0070, Bel-Art) and centrifuged at 300$g$ for 5 min at 4 °C. The supernatant was removed, and the nuclear pellet was resuspended in 200 µl of 0.5% BSA in PBS, supplemented with 0.2 U µl$^{-1}$ of Protector RNase inhibitors and filtered again through a 40-µm Flowmi Cell Strainer (H13680-0040, Bel-Art). Nuclei were quantified by Trypan blue staining using the Countess 3 automated cell counter (Thermo Fisher Scientific). Only Trypan blue-positive nuclei were counted (to exclude fat droplets). Each nucleus suspension was normalized to a concentration of 800 nuclei per µl using the resuspension buffer.

### PCR with reverse transcription
RNA was isolated from total thymus tissue using TRI reagent (T9424, Sigma-Aldrich). Thymus tissue was isolated from specimens of unspecified sex from juvenile brown-banded bamboo shark (*C. punctatum*)[13] and *foxn1*$^{+/+}$ and *foxn1*$^{-/-}$ adult zebrafish[56] on the *ikzf1*:EGFP background[43]. For each RNA extraction from zebrafish thymi, three organs were pooled and a total of three such pools were processed. RNA was quantified using the Qubit RNA HS Assay Kit (ThermoFisherScientific, Q32852) and the Qubit 4 Fluorometer (ThermoFisherScientific, Q33226). RNA quality was checked by determining the 18S/28S rRNA ratio using the Fragment Analyzer RNA Kit (ThermoScientific, DNF-471-0500) and the 5200 Fragment Analyzer System (ThermoScientific, M5310AA). cDNA was prepared using random hexamers and the SMARTScribe Reverse Transcriptase (Clontech).

The primers used for amplification from bamboo shark cDNA and the respective amplicon sizes are listed in Supplementary Table 3; the primers used for amplification from zebrafish cDNA and the respective amplicon sizes are listed in Supplementary Table 4. In the PCR reaction, primers were used at a concentration of 10 pmol µl$^{-1}$. For comparative analyses, the amounts of input cDNA were calibrated by comparison to zebrafish *ef1a* expression levels. Amplicons, independently generated for three thymic samples each per genotype, were pooled prior to gel electrophoresis. For gel source data, see Supplementary Fig. 8.

### Bulk RNA-seq of TECs
TECs derived from mouse thymus were sorted directly into TRI reagent (T9424, Sigma-Aldrich), while for zebrafish total thymus tissue was used. RNA isolation was performed according to standard protocols. RNA was quantified using the Qubit RNA HS Assay Kit (ThermoFisher-Scientific, Q32852) and the Qubit 4 Fluorometer (ThermoFisherScientific, Q33226). RNA quality was checked by determining the 18S/28S rRNA ratio using the Fragment Analyzer RNA Kit (ThermoScientific, DNF-471-0500) and the 5200 Fragment Analyzer System (ThermoScientific, M5310AA). For each zebrafish library, RNA from three animals was pooled and three such pools were sequenced. Libraries were prepared using the Ultra RNA Library Prep Kit (NEB). Samples were run on Illumina HiSeq 2500, HiSeq 3000 or NovaSeq 6000 instruments and sequenced to a depth of $10 \times 10^6$ to >$150 \times 10^6$ reads per sample. Transcriptomes were analysed on the Galaxy platform[59] using Trim Galore! version 0.4.3.1 (developed by Felix Krueger at the Babraham Institute), HISAT2 version 2.1.0[60] and featureCounts version 1.6.1.0[61].

### snRNA-seq of thymic tissue
The Chromium GEM-X Single Cell 3' v4 protocol (CG000731, Rev B) was followed starting from step 1.1 according to the manufacturer's guidelines. The Chromium GEM-X Single Cell 3' Kit v4 (PN-1000686) and the Chromium GEM-X Single Cell 3' Chip Kit v4 (PN-1000690) were used to process each sample. A total of 36.6 µl of normalized nucleus suspension (see 'Isolation of cell nuclei from thymus tissue') was added to the master mix to target a recovery of 20,000 nuclei. The ready GEM-X chip was loaded onto a 10x Genomics Chromium Xo instrument, and final libraries were completed as per 10x Genomics guidelines. Libraries were quantified using the Qubit High Sensitivity DNA Assay (Invitrogen, Q32851), and their molar concentrations were calculated on the basis of their size distribution using the Fragment Analyzer NGS 1–6,000 bp High Sensitivity DNA kit. Libraries were pooled, cleaned of adapter dimers and denatured according to Illumina guidelines. Libraries were sequenced in paired-end mode with a read length of $10 \times 100 \times 100 \times 10$ bp (i5 index, R1, R2, i7 index) on a NovaSeq 6000 instrument (Illumina), aiming for a minimum sequencing depth of 20,000 read pairs per nucleus.

### Analysis of snRNA-seq data
Counts were generated with 10x Genomics Cell Ranger (8.0.1) and background noise was reduced with CellBender (0.3.0, --fpr 0.01)[62]. Droplets were excluded if they exhibited low total UMI counts (<500), high proportions of mitochondrial counts (>5%), or abnormal numbers of detected genes (beyond 1 median absolute deviation (MAD) from the median) or complexity (ratio of detected genes over total count; beyond 3 MADs from the median). Doublets were removed with scDblFinder[63]. After initial dimensionality reduction and clustering of the data, each cell was scored for mimetic and background marker gene sets with AUCell[21]. Clusters predominantly comprising cells scoring highly for signatures compatible with expected contaminants (thymocytes and adipocytes for control and nude samples, respectively) were excluded from further analysis. The reduced dataset was clustered again. Parameters were selected to maximize the mean silhouette width. Subclustering was performed on clusters selected manually based on their mean silhouette width, again performing parameter sweeps and selecting parameters based on the total within cluster sum of squares and mean silhouette widths of subclusters. The final clusters were annotated by investigation of dominant signature scores and marker genes.

### Assessment of mimetic signatures in bulk RNA-seq data
Bulk RNA-seq data from all samples were processed jointly. Genes were included if their counts exceeded 5 in at least 50% of samples. Differential expression of genes between samples was assessed with voom[64] and limma with treat(., lfc = log2(1.2), robust = TRUE)[65,66] to generate

the *t*-statistic. Relative enrichment of gene signatures was determined with the competitive gene set test camera[25] with cameraPR, employing the *t*-statistic as output by treat as the ranking statistic. The output of camera is a two-sided and Benjamini–Hochberg adjusted *P* value. Results are reported as $\log_{10}$(adj. *P*), conditionally multiplied by −1 for signatures with an upwards direction of change. Although we were mainly interested in TEC signatures[2,20], we also collected background signatures of unrelated cell types from PanglaoDB[67], Tabula Muris[68] (bladder, spleen, kidney, liver, marrow, muscle, lung, non-myeloid brain, from FACS) and MSigDB[69,70] (M8: mouse cell-type signature gene sets, except for Tabula Muris Senis signatures). From the collection of signatures, those that comprised up to 10% less or 10% more genes than the smallest and largest TEC or mimetic signature, respectively, were retained as background signatures (see Supplementary Fig. 1). To investigate mouse-derived signatures in RNA-seq samples from *D. rerio*, we performed orthology transfer using data from the Ensembl database (release 112)[71,72], allowing many-to-many relationships.

### Identification of mimetic cells in scRNA-seq data

Signature gene sets for mimetic cells were derived from single-cell differential gene expression results from Supplementary Table 2 in Michelson et al.[2]. The gene lists were filtered (adjusted *P* < 0.01, |$\log_2$(FC)| >1, expressed in at least 10% of cells) and used as input to AUCell[21] to score signatures in each cell. Based on the resulting histograms of areas under the curve (AUCs), thresholds were defined manually for each mimetic signature (see Extended Data Fig. 1a). A cell that met thresholds for multiple signatures was disambiguated by ordering all eligible signatures and choosing the highest-ranked one, first by the cell's rank within the signature AUCs (that is, a signature is ranked higher if the cell's position on the histogram is further to the right), and second (in case of ties) by the *z*-score standardized AUC. The signatures 'Tuft1' and 'Tuft2', as well as 'Skin_basal' and 'Skin_keratinized' led to the identification of an overlapping set of cells which were therefore collapsed into unified Tuft and Skin populations (Extended Data Fig. 1b). Using the final signatures, mimetic cells were identified in the uniform manifold approximation and projection (UMAP) graphs calculated from single-cell datasets[20].

Shifts in population composition were evaluated with scCODA[24]. The model was run multiple times, using each of the major populations (early and postnatal progenitors, mTEC, cTEC, Aire-stage and unassigned) as the reference. Detected changes were minimal for the 'unassigned' population and it was therefore used as the final reference. A population was deemed 'overall credibly changed' if more than half the models detected a credible change.

### Lineage tracing of mimetic cell lineages

CRISPR–Cas9 barcodes for single cells[20] were binarized (barcode present or absent) and a logistic regression model was fitted with fixed covariates 'signature' and 'sample'. A *P* value for the 'signature' term was determined by likelihood ratio test with a reduced model. Proportions of *Foxn1*-expressing cells between populations were compared with a binomial test, implemented in the findMarkers function from the scran package[73].

### Determination of fate probabilities in scRNA-seq data

scRNA-seq data from barcoded and unbarcoded samples from three time points (embryo, newborn, 4 week) were integrated with fastMNN[74]. The integrated dataset was re-clustered and clusters with expression of *Pth*/*Chga* (ectopic parathyroid) and *Cd3*/*Cd4*/*Cd8a* (thymocytes) were excluded. Diffusion pseudotime was calculated, starting from the cell with the highest AUC for the early progenitor signature in the embryonic sample[75]. The pseudotime and integrated representation of the data were used for generation of a PseudotimeKernel and ConnectivityKernel with CellRank[29,30], respectively. Both kernels were combined and analysed with a GPPCA estimator[76] to identify between 10 and 16 macrostates

(gppca.fit(n_states = [10,16])). The optimal number of macrostates based on the *minChi* criterion[76] was 12. Macrostates were manually labelled according to their composition of annotated cell types and defined as initial (early progenitors) or terminal (Aire-stage and mimetic populations) states. Intermediate states were not retained. Fate probabilities were calculated with the remaining macrostates and the overlap of cells exceeding fate probability thresholds in pairs of fates was quantified by the Jaccard index. Because the automatically determined macrostates did not reflect all mimetic populations, we additionally ran CellRank after manually identifying terminal cells, picking representatives with the highest signature AUC for each population of interest. Fate probabilities and Jaccard indices were also calculated for these data.

### Data handling and statistics

Box plots encapsulate the first to third quartile, a line indicates the median. Whiskers extend to the furthest point with a distance of up to 1.5 times the interquartile range from the boxes. For the data analyses presented in Figs. 2a,b and 3e and Extended Data Figs. 6 and 8a, differences in means between groups were compared by ANOVA. Separate ANOVAs were conducted for each dependent variable. The resulting *P* values were corrected for multiple testing with the conservative Bonferroni method. For tests with adjusted *P* < 0.05, we rejected the null hypothesis of equal means and performed pairwise two-sided *t*-tests between all (Figs. 2a,b and 3e and Extended Data Fig. 6a) groups, or between selected groups (Extended Data Fig. 8, $F_2$ samples only). *P* values from pairwise *t*-tests were corrected for multiple testing with the conservative Bonferroni method. For Extended Data Figs. 2d and 4d, groups were compared via a two-sided *t*-test without (Extended Data Fig. 2d) or with (Extended Data Fig. 4d) Welch correction (GraphPad Prism 9.5.1). For LOESS (Fig. 3a), we tested span parameters between 0.4 and 0.9 in increments of 0.01 with tenfold cross-validation. The minimal mean squared error was achieved with a span of 0.53. The numbers of biological replicates are indicated in the figure panels or figure legends.

### Reporting summary

Further information on research design is available in the Nature Portfolio Reporting Summary linked to this article.

## Data availability

All data supporting the findings of this study are available within the Article and its Supplementary Information. Sequence data are deposited at the Gene Expression Omnibus (GEO). Bulk RNA-seq data are available under accession numbers GSE272144 (mouse), GSE272063 (mouse) and GSE272064 (zebrafish). Data under GSE272144 were previously reported[38] and reanalysed here. Data from mouse snRNA-seq are available under accession number GSE288957. Background signature gene sets are available from PanglaoDB (https://panglaodb.se/markers.html), the Tabula Muris repository (https://github.com/czbiohub-sf/tabula-muris/tree/dedd8352d4348150e199162f966f7442976acdd3/22_markers) and MSigDB (https://www.gsea-msigdb.org/gsea/msigdb/mouse/genesets.jsp?collection=M8). Source data are provided with this paper.

## Code availability

Raw data and the code to generate the figures for this manuscript can be found on Github (https://github.com/osthomas/mimetics_evodevo). Analyses were run in conda (23.3.1)/mamba (1.4.2) environments with the following specifications: *R*: r-base=4.2.3, r-here=1.0.1, r-tidyverse=1.3.2, r-ggplot2=3.5.1, r-remotes=2.4.2, r-devtools=2.4.5, r-matrix=1.6_1, r-matrixstats=1.0.0, r-patchwork=1.2.0, r-geomtextpath=0.1.4, r-dbplyr<=2.3.4, r-desctools=0.99.51, r-writexl=1.5.0, bioconductor-biomart=2.54.0,

bioconductor-aucell=1.20.1, bioconductor-scran=1.26.0, bioconductor-scater=1.26.0, bioconductor-batchelor=1.14.0, bioconductor-complexheatmap=2.14.0 *cellbender*: python=3.7; *pip*: cellbender=0.3.0 *sccoda*: python=3.12.4; *pip*: pertpy=0.7.0, jax=0.4.30 *scrnaseq*: python=3.11.9, r-base=4.2.3, r-reticulate=1.38.0, quarto=1.5.57, r-tidyverse=1.3.2, r-ggplot2=3.5.1, r-remotes=2.4.2, r-devtools=2.4.5, r-patchwork=1.2.0, r-geomtextpath=0.1.4, r-matrix=1.6_1, r-matrixstats=1.0.0, r-hdf5r=1.3.10, bioconductor-aucell=1.20.1, bioconductor-scran=1.26.0, bioconductor-scater=1.26.0, bioconductor-batchelor=1.14.0, bioconductor-scdblfinder=1.12.0, bioconductor-complexheatmap=2.14.0, bioconductor-zellkonverter= 1.8.0, anndata=0.11.1, scanpy=1.10.3, cellrank=2.0.6; *pip*: igraph=0.11.8, leidenalg=0.10.2 *zellkonverter*: r-base=4.2.3, r-tidyverse=1.3.2, bioconductor-scran=1.26.0, bioconductor-scater=1.26.0, bioconductor-zellkonverter=1.8.0.

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

**Acknowledgements** The authors thank B. Kanzler for help with generation of transgenic mice, J. Swann for help with nude mouse tissue dissection, M. Kauka for *S. canicula* specimens and F. Guillemot for the *Ascl1* mouse strain. This study was supported by the Max Planck Society.

**Author contributions** A.N. and T.B. conceived the study. A.N. and O.S.T. developed the methods used in this study. A.N., O.S.T., G.Z., D.N., L.A. and B.K. carried out the research. A.N. and T.B. wrote the manuscript with input from all authors.

**Funding** Open access funding provided by Max Planck Society.

**Competing interests** The authors declare no competing interests.

**Additional information**
**Correspondence and requests for materials** should be addressed to Thomas Boehm.

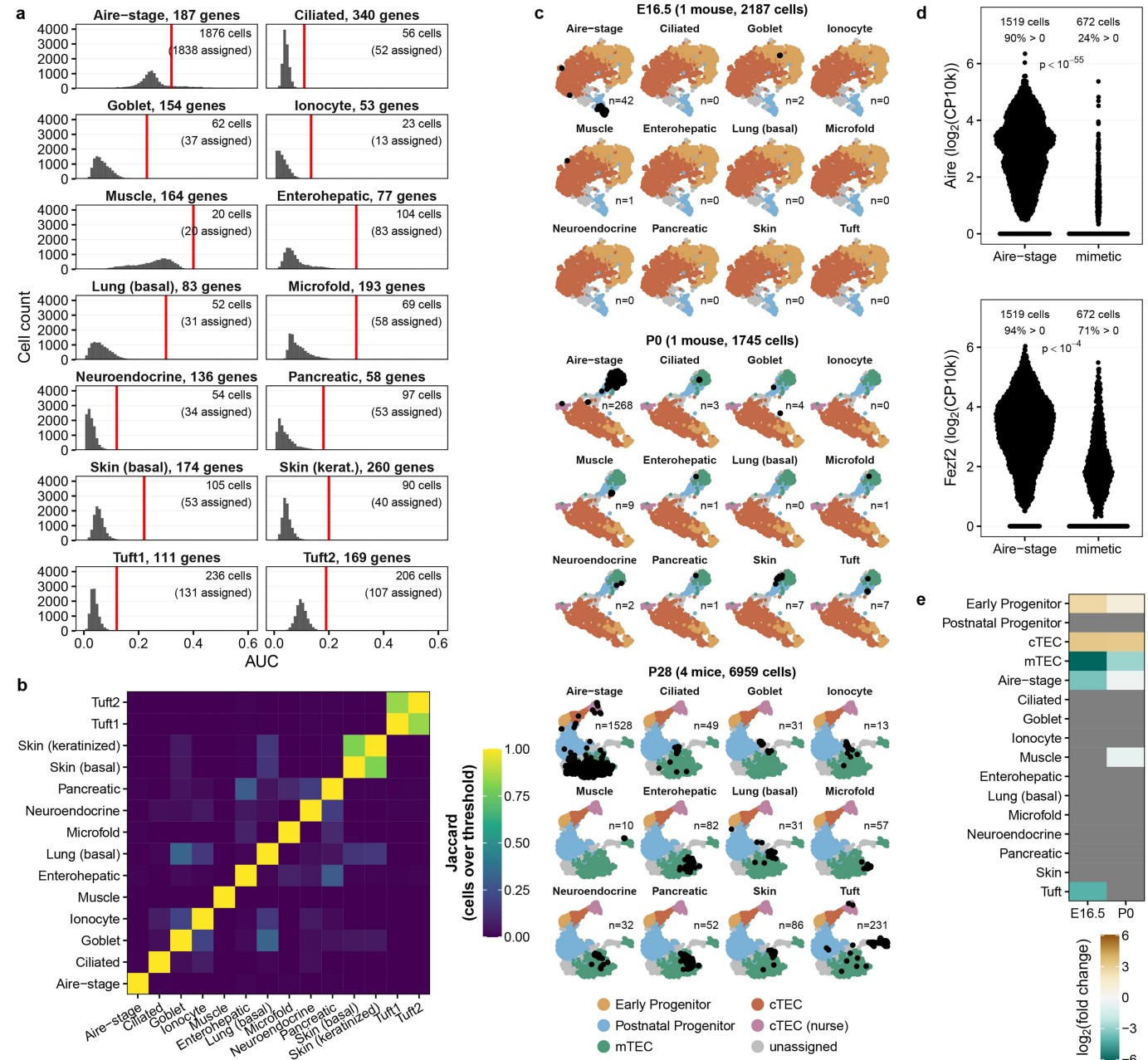

**Extended Data Fig. 1 | Identification of mimetic cells in scRNAseq data.**
**a**, All cells were scored for all listed signatures, resulting in one area under the curve (AUC) per cell and signature. The resulting histogram of AUCs for each signature is shown. Red lines indicate chosen AUC thresholds. Cell numbers indicate cells above the threshold value, whereas assigned cells refers to final assignments after resolution of ambiguities. **b**, Jaccard index for sets of cells meeting the threshold for each pair of signatures. In our data set, skin (basal)/ skin (keratinized) and tuft1/tuft2[2] led to the identification of an overlapping set of cells. Thus, they were collapsed into skin and tuft populations, respectively. **c**, UMAPs of scRNAseq data. Data and overall population labels (colored) were reported previously[20]. The counts for each mimetic population are given per panel, and corresponding cells are highlighted in black. **d**, Overall *Aire* and *Fezf2* expression levels ($\log_2$ of counts per 10,000 reads) in Aire-stage and mimetic cells from scRNAseq data (n = 4 mice; P28). The proportions of cells with detectable gene expression is indicated. $p(Aire) = 3.6 \times 10^{-56}$; $p(Fezf2) = 2.8 \times 10^{-5}$ (two-sided binomial test, Benjamini-Hochberg adjusted). **e**, Results of differential abundance analysis with scCODA[24]. Population $\log_2$ fold changes are indicated. Gray boxes indicate no detectable change. The "unassigned" population was used as the reference population.

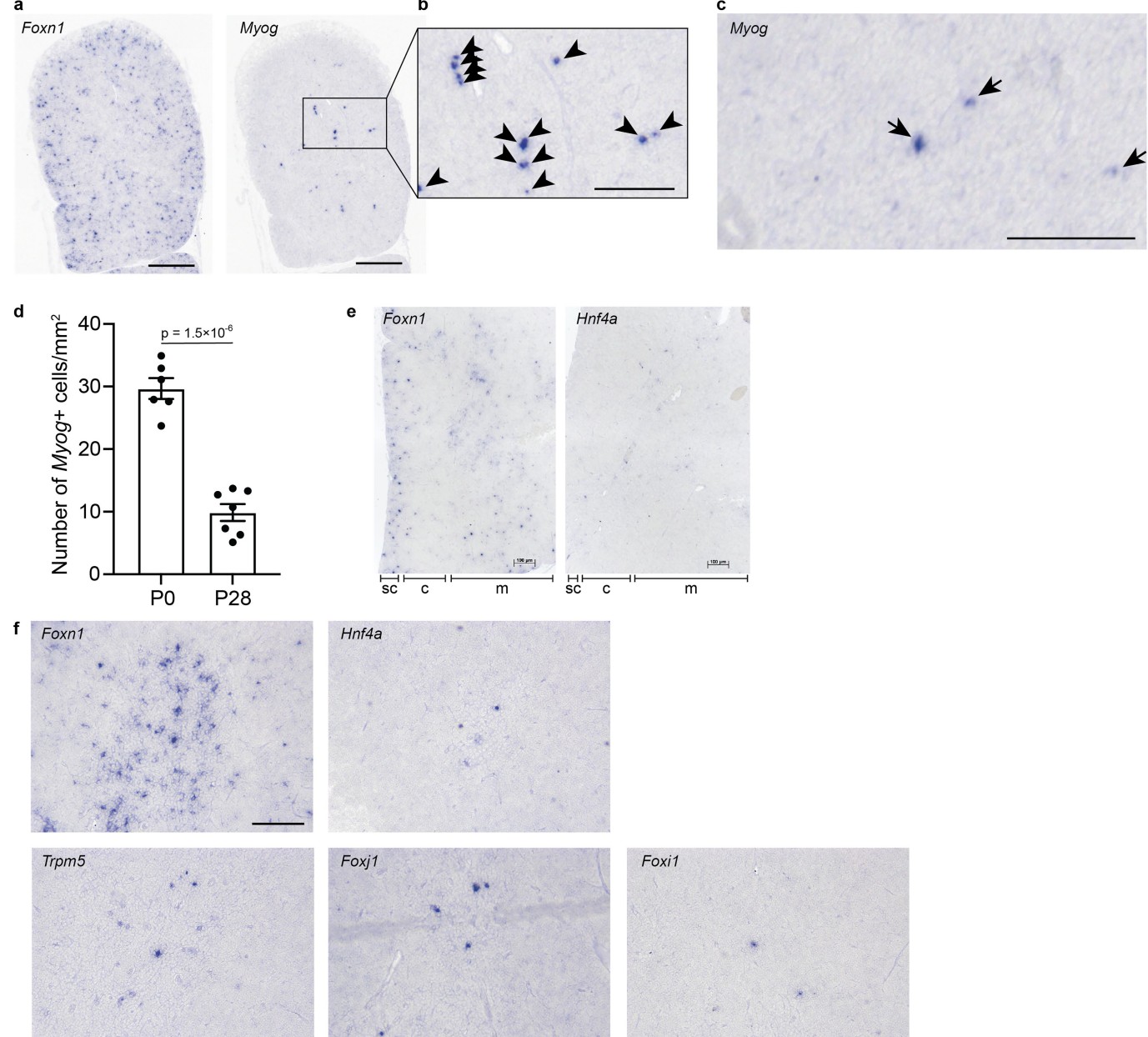

**Extended Data Fig. 2 | Expression patterns of TEC-specific genes in the mouse thymus. a**, Micrographs of P0 thymus sections hybridized with the indicated probes. **b**, **c**, Higher magnification views of the *Myog* hybridization pattern at P0 (b) and P28 (c). **d**, Enumeration of *Myog*-positive cells in thymus sections of P0 (n = 6) and P28 (n = 7) mice. Each data point represents a different thymus section. Data are shown as mean ± SD; $p = 1.5 \times 10^{-6}$, two-sided *t* test. Results in **a**-**d** are representative of n = 2 mice of each time point with similar results. **e**, RNA in situ hybridization of consecutive sections of *Foxn1*[+/+] thymi at

P28, hybridized with probes specific for *Foxn1* (pan TEC marker) and *Hnf4a*, a marker of the enterohepatic mimetic lineage. The relevant tissue compartments are indicated (sc, subcapsular region; c, cortex; m, medulla). Note that the *Hnf4a* signal is confined to the medullary region. **f**, Higher magnification views of the medullary regions of P28 thymi hybridized with the indicated probes. Results in **e**, **f** are representative of n = 3 mice with similar results. Scale bars: **a**, 0.2 mm; **b**, **c**, **e**, **f**, 0.1 mm.

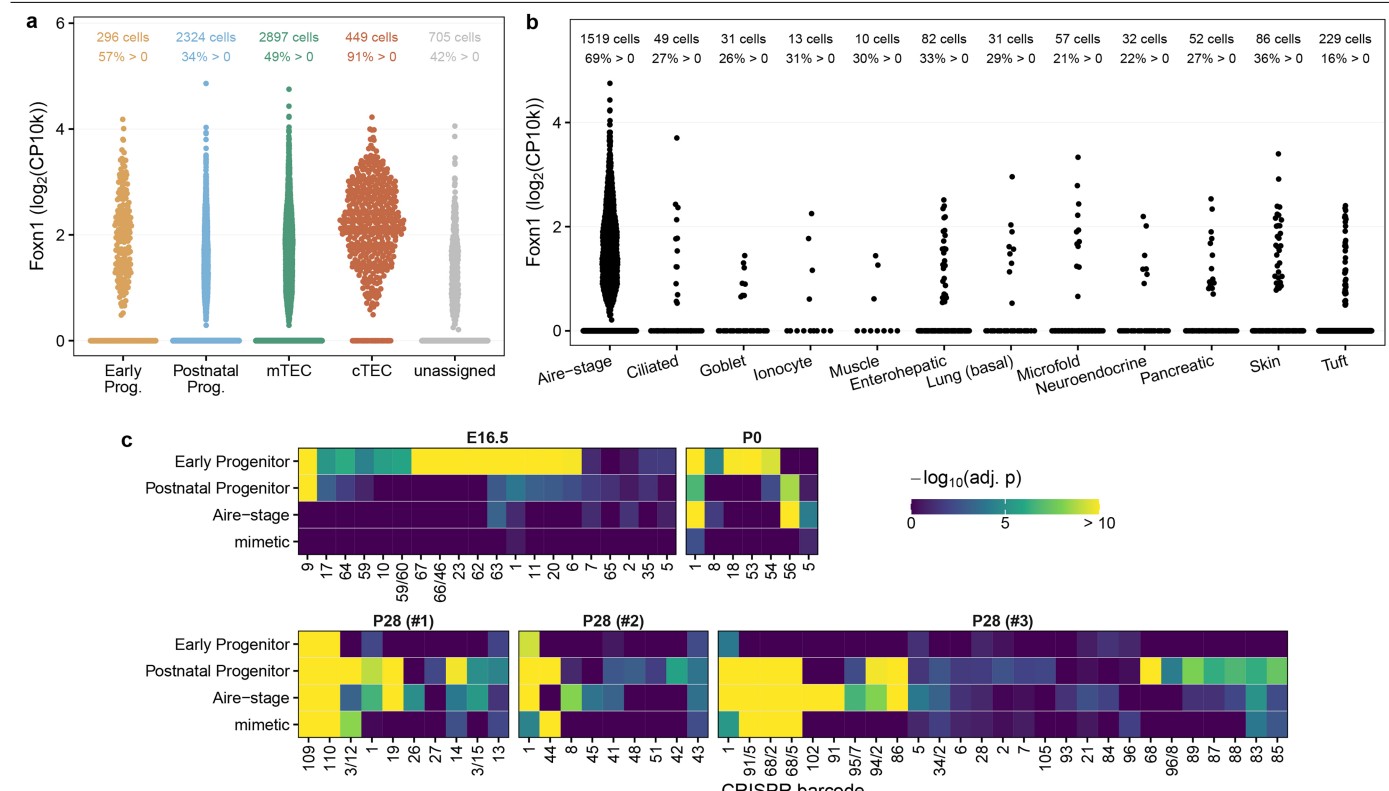

**Extended Data Fig. 3 | *Foxn1* expression levels in TEC populations.**
**a, b**, Expression levels in canonical TEC populations (**a**), and Aire-stage and mimetic populations (**b**); data combined from 4 mice (P28). The proportions of cells with detectable *Foxn1* are indicated. **c**, Sampling probabilities of individual informative CRISPR/Cas9-induced barcodes for the indicated populations and time points, calculated as described previously[20]. A barcode was deemed informative if $p \leq 0.05$ for any population.

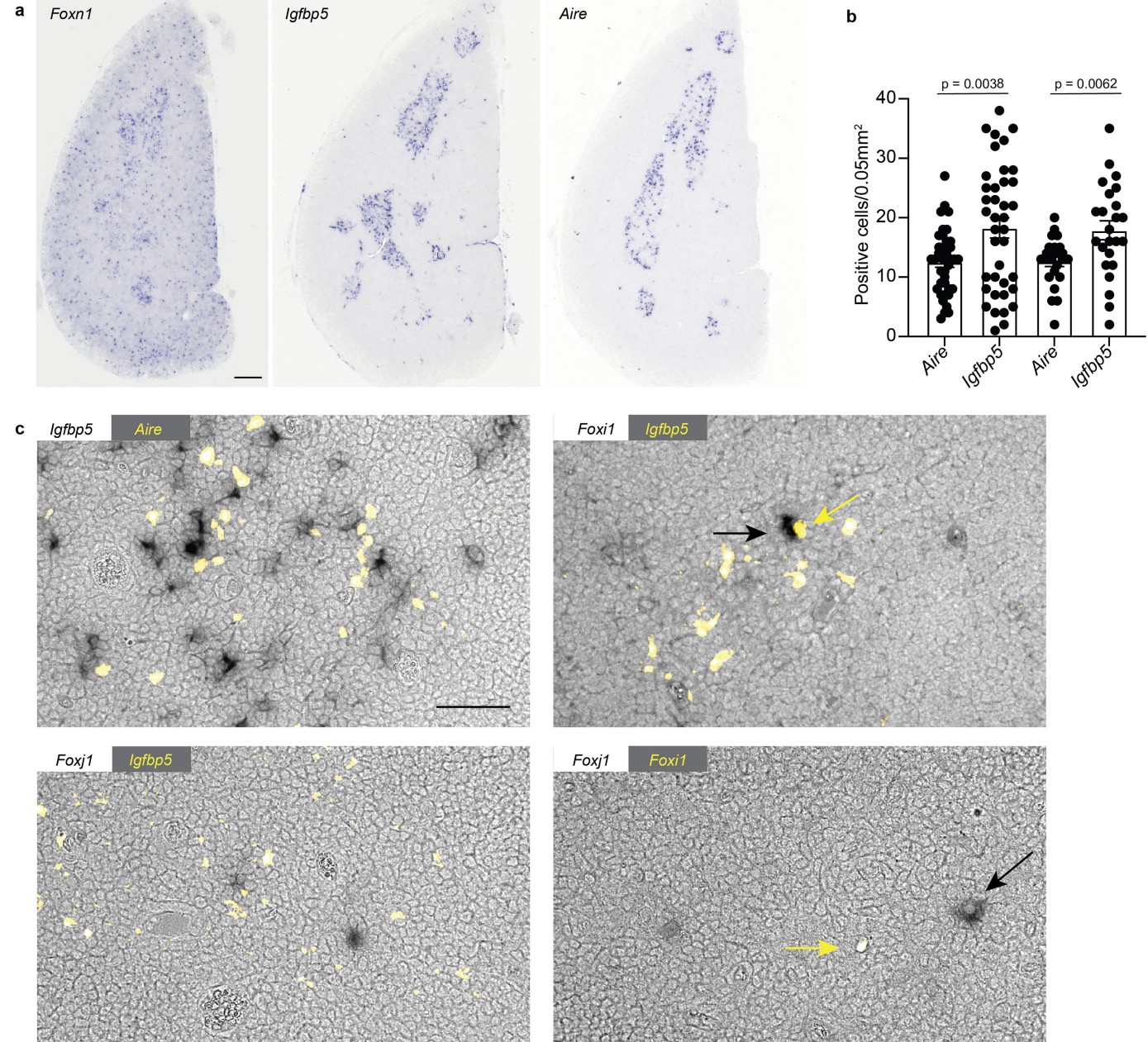

**Extended Data Fig. 4 | Characterization of TEC populations in the P28 mouse thymus by RNA in situ hybridization. a**, Hybridization patterns obtained from hybridization with the indicated probes. Note the strong signals for *Foxn1* in the subcapsular and medullary regions; the *Igfbp5*- and *Aire*-expressing cells are located in medullary areas. Scale bar, 0.2 mm. Panels are representative of n = 3 mice with similar results. **b**, Enumeration of *Igfbp5*- and *Aire*-expressing cells in the thymic medulla. Each data point represents a different section; the data from the two thymic lobes of one mouse were combined; results are representative of n = 2 mice. The data for the left pair of *Aire*/*Igfbp5* columns (both, n = 39) were determined using a Cy5-labeled *Aire* probe and a digoxigenin-labelled *Igfbp5* probe revealed by chromogenic detection; the data for the right pair of *Aire*/*Igfbp5* columns (both, n = 24) were determined using a Cy5-labeled *Aire* probe and a Cy3-labelled *Igfbp5* probe. Data are shown as mean ± SD; *p*(left pair)=0.0038; *p*(right pair)=0.0062; two-tailed *t* test with Welch correction. **c**, Double hybridization patterns for the indicated probe combinations. The dark signals emanate from digoxigenin-labelled probes; the yellow signals emanate from Cy5-labelled probes. The panels are representative of n = 5 sections each; similar results were obtained for n = 2 mice. Scale bar, 0.1 mm.

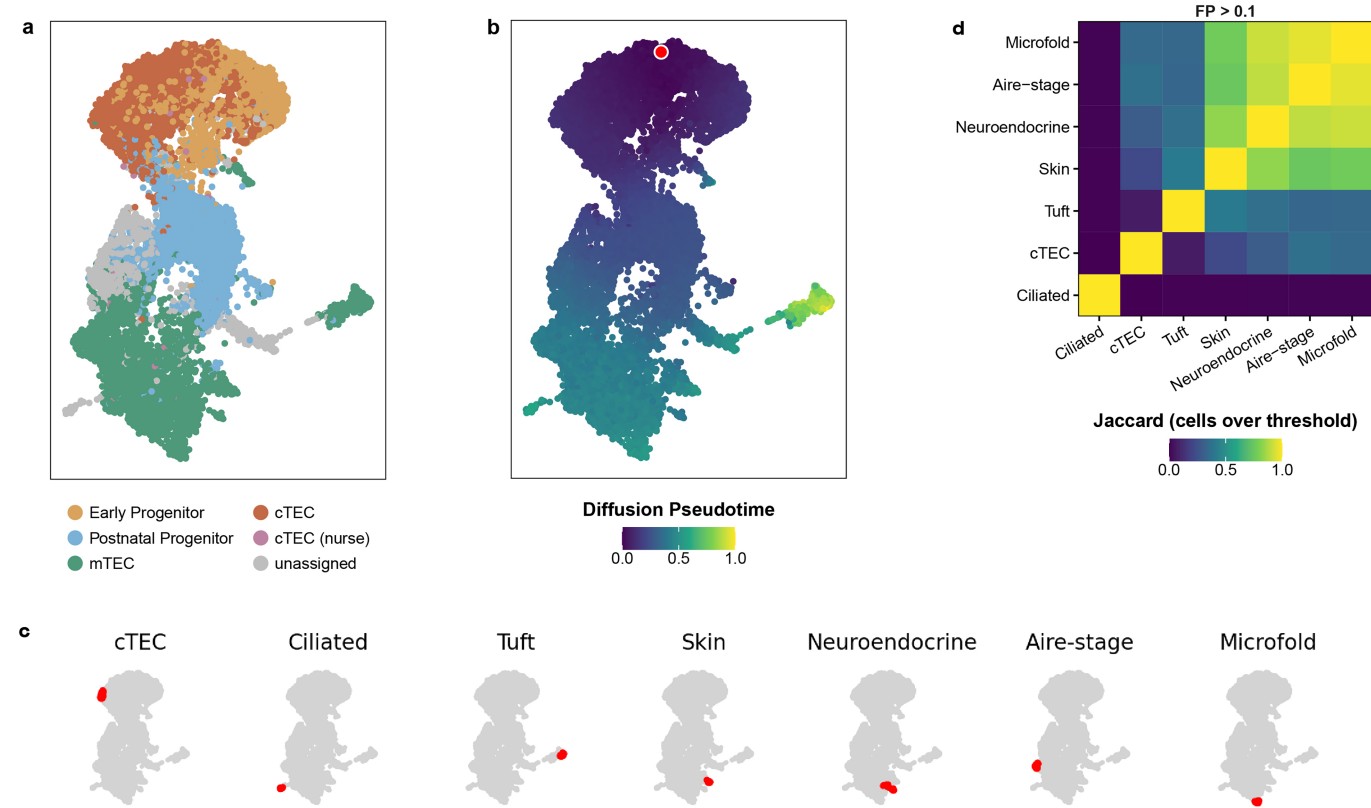

**Extended Data Fig. 5 | Fate trajectories of mouse TECs. a**, Population labels from[20] overlaid on the UMAP embedding obtained after integration of data from embryonic, newborn and 4 week old mice. **b**, Diffusion pseudotime overlaid on the integrated UMAP. The starting cell, an early progenitor from the embryo, is highlighted in red. **c**, Location of cells comprising macrostates identified by CellRank which were assigned as terminal states. These macrostates were considered for calculation of fate probabilities. **d**, Jaccard indices of sets of cells for pairs of fates. The set of cells for each fate comprises cells meeting the fate probability threshold of 0.1.

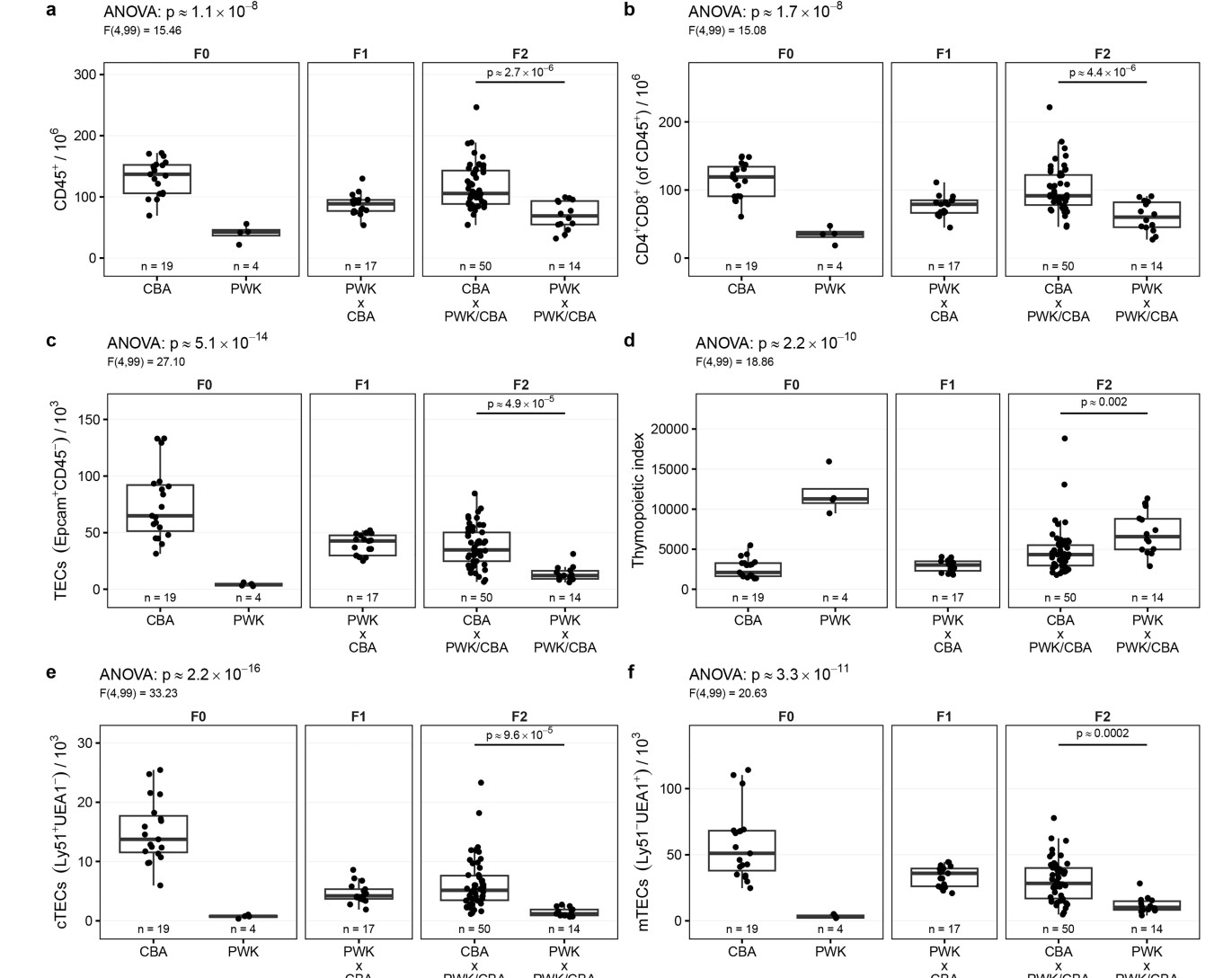

**Extended Data Fig. 6 | Characterization of thymic populations in CBA and PWK mice, CBAxPWK F1 hybrids, and two reciprocal backcrosses (F2).** **a**, Absolute number of CD45$^+$ haematopietic cells. **b**, Absolute numbers of CD4/CD8-double positive thymocytes. **c**, Absolute numbers of EpCAM$^+$CD45$^-$ TECs. **d**, Thymopoietic index, calculated as the ratio of CD4/CD8-double positive thymocytes and TECs. **e**, Absolute numbers of EpCAM$^+$CD45$^-$ Ly51$^+$UEA1$^-$ cTECs.

**f**, Absolute numbers of EpCAM$^+$CD45$^-$ Ly51$^-$UEA1$^+$ mTECs. **a-f**, n as indicated in the panels, each data point shows one thymus explant from one animal. *p* values between groups were derived from pairwise two-sided t tests with Bonferroni correction after significant (*p* < 0.05) ANOVA result. Boxplots encapsulate the first to third quartile, a line indicates the median. Whiskers extend to the furthest point with a distance of up to 1.5 times the interquartile range from the boxes.

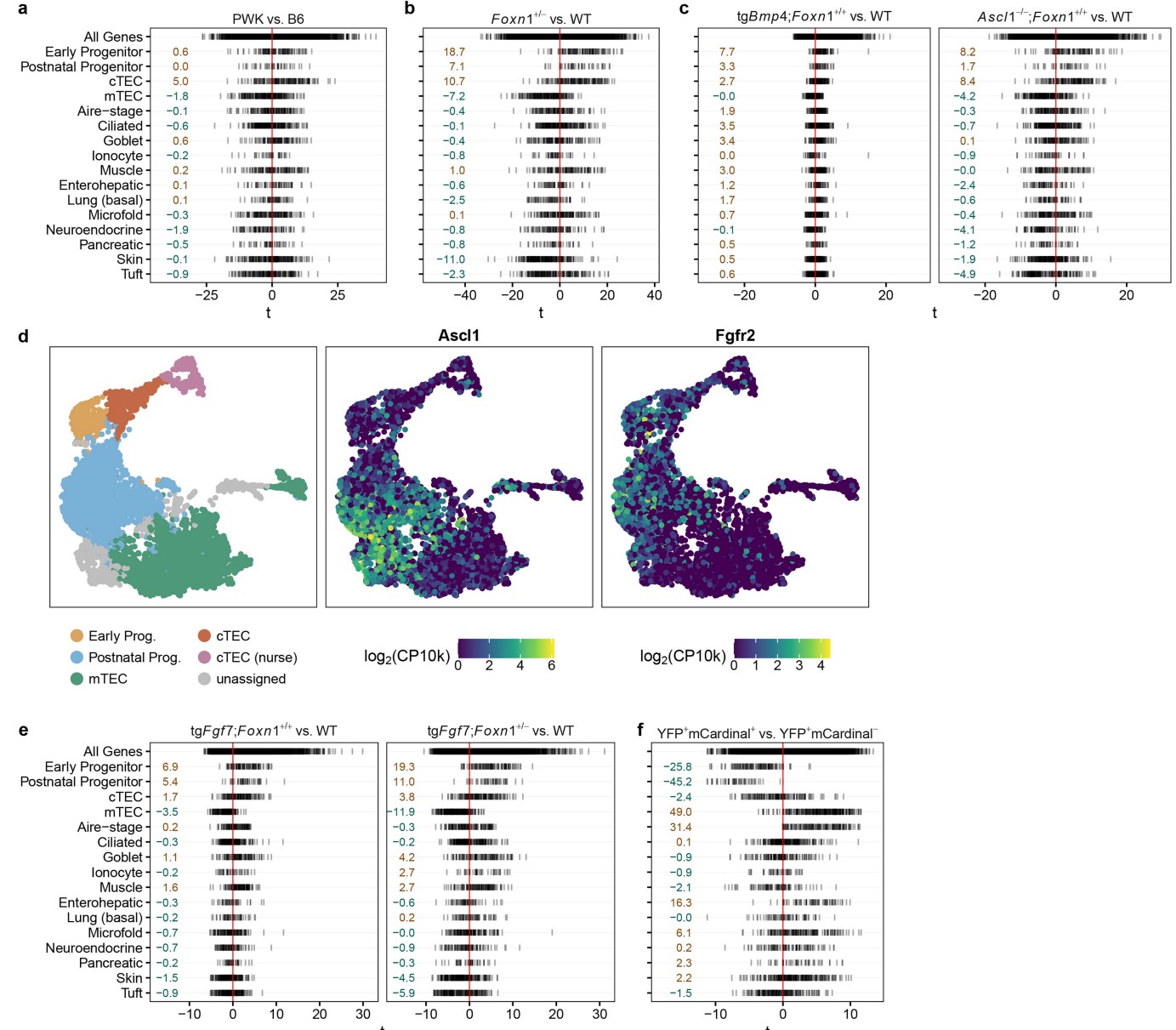

**Extended Data Fig. 7 | Comparison of canonical TEC and mimetic signatures in bulk RNAseq data.** Visualization of expression changes between **a**, PWK and C57BL/6 mouse strains, **b**, *Foxn1*[+/−] and *Foxn1*[+/+] strains, **c**, between *Foxn1:Bmp4* transgenic and nontransgenic mice (left panel)), and between TEC-specific *Ascl1*-deficient (*Ascl1*[fl/fl]; *Foxn1:Cre*) and control (*Ascl1*[+/+]; *Foxn1:Cre*) mice (right panel). In **a**, **b**, **c**, and **e**, **f**, each line represents a gene of the indicated signature and its position on the x axis shows the value of the *t* statistic derived from differential expression analysis. Numeric values listed in the left column represent $\log_{10}(\text{adj. } p)$ from enrichment analysis with *camera*[25]. Log transformed *p* values

of upregulated sets were multiplied with −1 so that positive values indicate upwards directionality, and negative values indicate downwards directionality, respectively. **d**, Expression pattern of the indicated genes visualized on the UMAP of purified TECs at P28 (see Extended Data Fig. 1c). **e**, Visualization of expression changes between *Foxn1*[+/+]; *Foxn1:Fgf7* and *Foxn1*[+/+] (left panel), and *Foxn1*[+/−]; *Foxn1:Fgf7* and *Foxn1*[+/+] (right panel). **f**, Expression changes and enrichment of TEC signatures from bulk RNAseq of purified mCardinal-positive and mCardinal-negative populations.

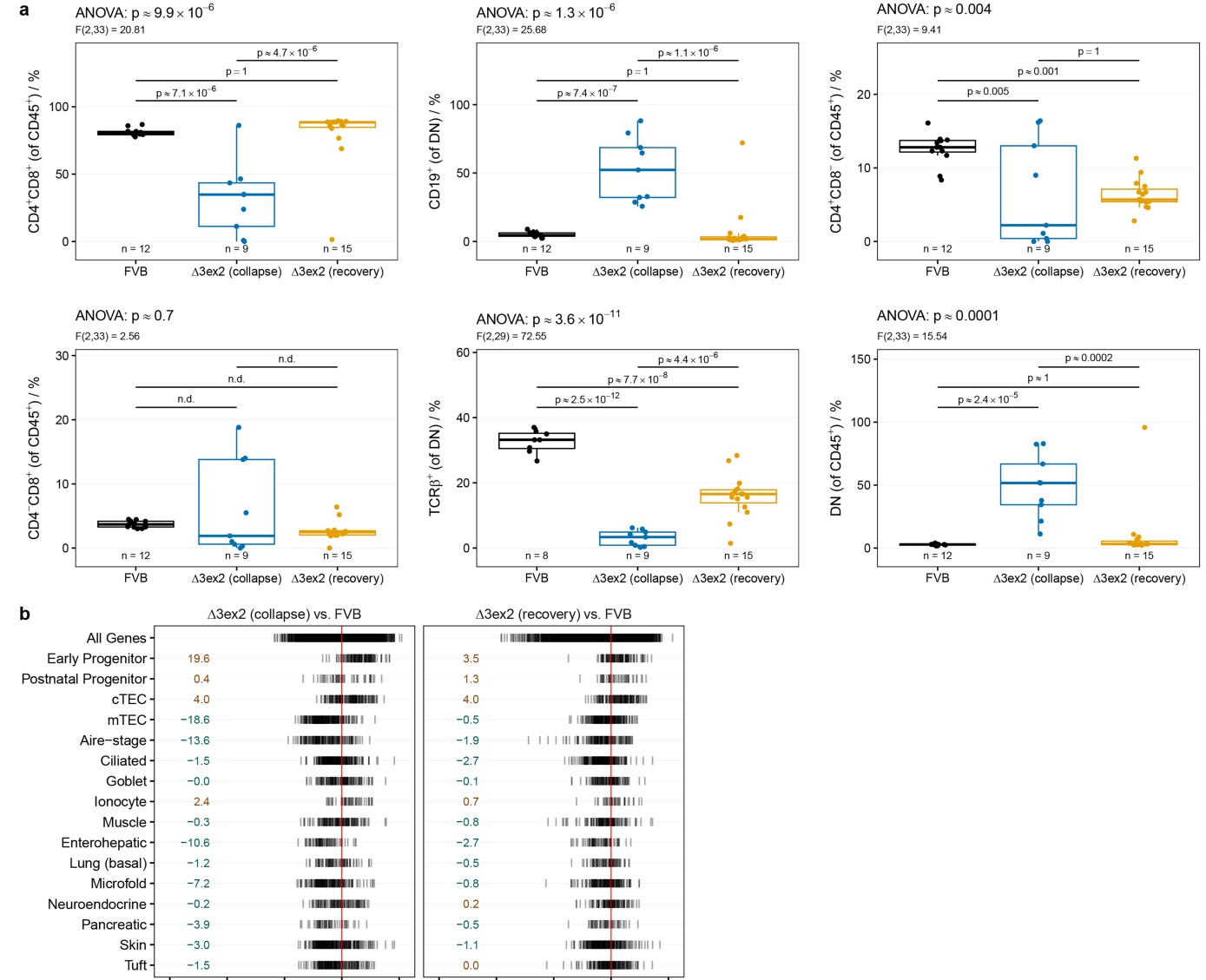

**Extended Data Fig. 8 | Characterization of thymic cellularity stratified according to total transcriptome of purified TECs. a**, Percentage of indicated haematopoietic cell types among thymocytes in non-transgenic control mice (FVB) and two groups of transgenic *Foxn1:Foxn1*$^{\Delta3ex2}$; *Foxn1*$^{-/-}$ mice assigned to the collapse and recovery phases. n as indicated in the panels, each data point shows one thymus explant from one animal. *p* values between groups were derived from pairwise two-sided *t* tests with Bonferroni correction after significant (*p* < 0.05) ANOVA result. Boxplots encapsulate the first to third quartile, a line indicates the median. Whiskers extend to the furthest point with a distance of up to 1.5 times the interquartile range from the boxes. **b**, Visualization of expression changes between *Foxn1:Foxn1*$^{\Delta3ex2}$; *Foxn1*$^{-/-}$ mice and non-transgenic wildtype mice stratified into collapse (left panel) and recovery phases (right panels). Each line represents a gene of the indicated signature and its position on the x axis shows the value of the *t* statistic derived from differential expression analysis. Numeric values listed in the left column represent log$_{10}$(adj. *p*) from enrichment analysis with *camera*[25]. Log transformed *p* values of upregulated sets were multiplied with −1 so that positive values indicate upwards directionality, and negative values indicate downwards directionality, respectively.

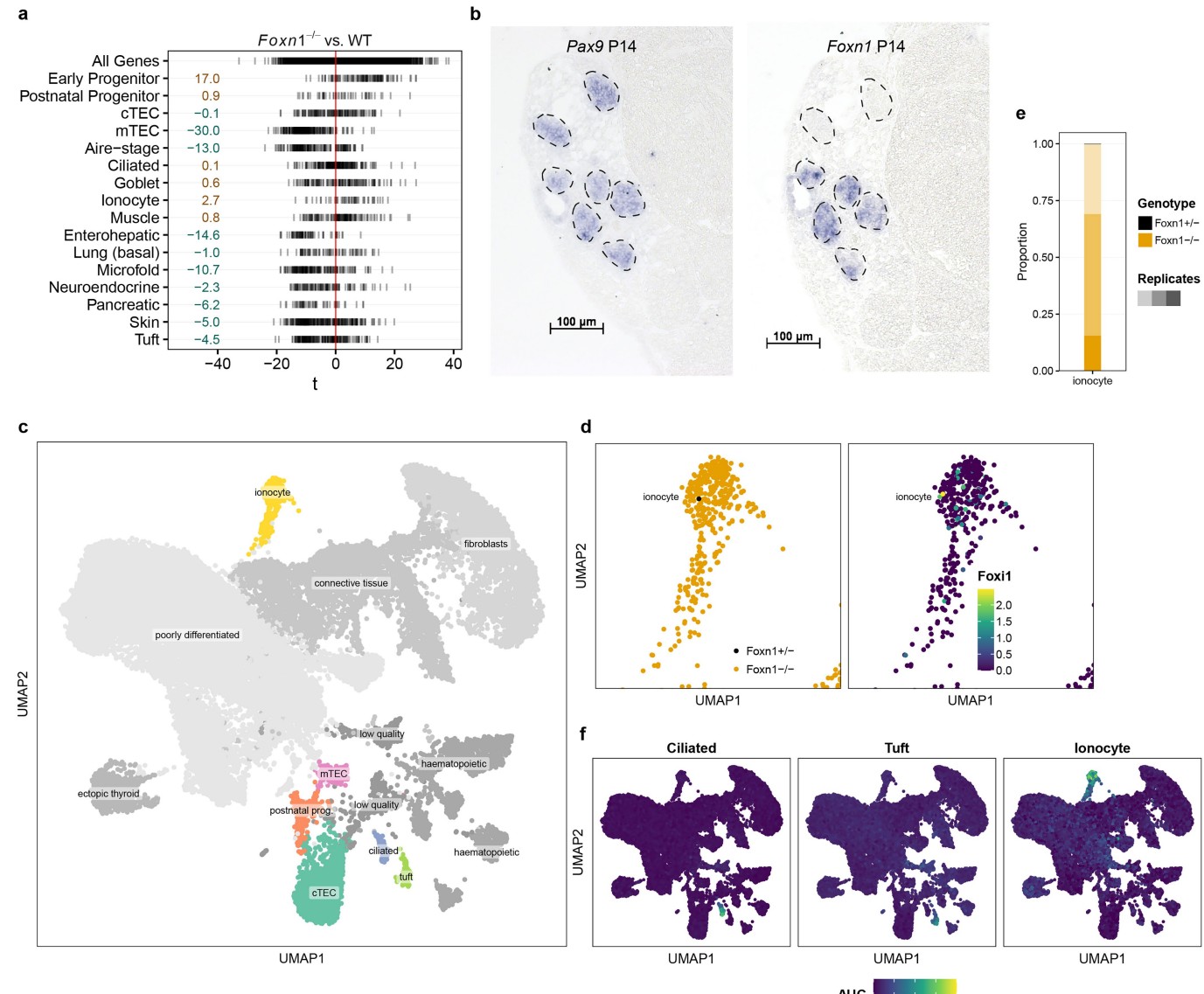

**Extended Data Fig. 9 | Characterization of the *Foxn1*-deficient thymic rudiment. a**, Visualization of expression changes between *Foxn1*⁻/⁻ mice and non-transgenic wildtype mice. Each line represents a gene of the indicated signature and its position on the x axis shows the value of the *t* statistic derived from differential expression analysis. Numeric values listed in the left column represent $\log_{10}$(adj. *p*) from enrichment analysis with *camera*[25]. Log transformed *p* values of upregulated sets were multiplied with −1 so that positive values indicate upwards directionality, and negative values indicate downwards directionality, respectively. **b**, RNA in situ hybridization of consecutive sections developed with the pharyngeal marker gene *Pax9* and the TEC-specific marker gene *Foxn1* at P14. Note that some patches of the *Pax9*-positive epithelium lack

*Foxn1* expression, indicating cellular heterogeneity of the *Foxn1*⁻/⁻ epithelium; scale bars, 0.1 mm. Representative for n = 4 mice with similar results. **c**, snRNAseq data from nuclei of thymus tissue from *Foxn1*⁺/⁻ (n = 3) and *Foxn1*⁻/⁻ (n = 6 mice, pooled in n = 3 samples of two animals each). UMAP of all nuclei after data processing, clustering and annotation. Broad cluster identities are labelled. **d**, Zoomed in view on the ionocyte cluster from **c**; each dot represents a cell, with the genotype of origin indicated by colour (left panel). Expression levels for *Foxi1* (right panel). Expression values are $\log_2$(normalized counts). **e**, Contributions of individual samples to clusters from **d**. Genotypes are coloured and replicates are are identified by different colour hues. **f**, Signature scores (AUCs[21]) for the indicated mimetic cell signatures in the UMAP of **c**.

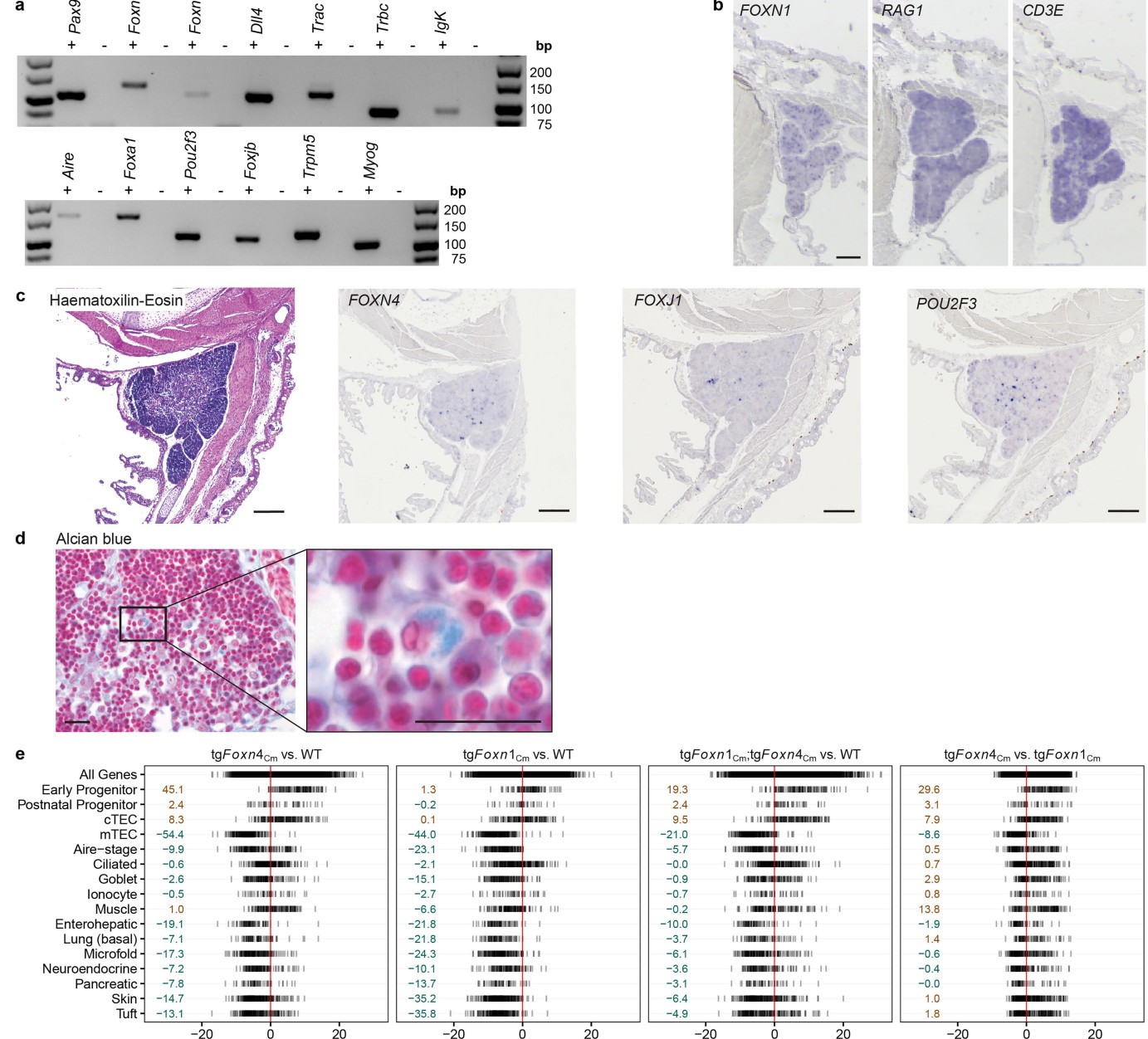

**Extended Data Fig. 10 | Analysis of thymopoiesis in cartilaginous fishes.**
**a**, Expressed genes in the thymus of the brown-banded bamboo shark
(*C. punctatum*); minus-RT control reactions (–) are shown in parallel; size
markers are given in base pairs (bp). For gel source data, see Supplementary
Fig. 8. **b**, Characterization of the thymus in *S. canicula*. Micrographs of the
thymus after RNA in situ hybridization with the indicated probes. Cells
expressing the indicated genes turn blue. **c**, Micrograph of the thymus region.
Haematoxilin-eosin staining of histological section is shown on the left;
micrographs of the thymus after RNA in situ hybridization with the indicated

probes are shown in the other panels. **d**, Thymus tissue section stained with
alcian blue to identify mucus-producing cells, highlighted in the inset. For **a-d**,
results are representative of n = 2 animals with similar results. **e**, Signature
enrichment analysis of mouse *Foxn1*[–/–] mutants expressing the *C. milii Foxn1*
and *Foxn4* genes under the control of the mouse *Foxn1* promotor compared
against corresponding wildtype controls. Log transformed *p* values of
upregulated sets were multiplied with −1 so that positive values indicate
upwards directionality, and negative values indicate downwards directionality,
respectively. Scale bars in **b**, **c** represent 0.2 mm; for **d**, 0.05 mm.

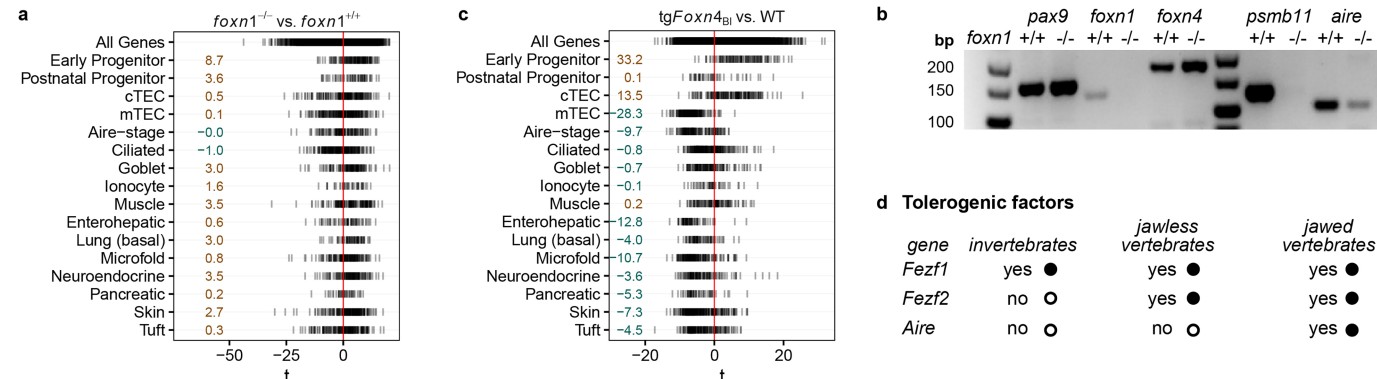

**Extended Data Fig. 11 | Evolutionary aspects of mimetic cell development.**
**a**, Expression changes in thymi of *foxn1*⁻/⁻ zebrafish compared to *foxn1*⁺/⁺ controls. Log transformed *p* values of upregulated sets were multiplied with −1 so that positive values indicate upward directionality, and negative values indicate downwards directionality, respectively. **b**, RT-PCR analysis of indicated genes in zebrafish *foxn1*⁺/⁺ and *foxn1*⁻/⁻ thymic tissues. Representative results for n = 6 animals of each genotype with similar results. Size markers are given in base pairs (bp). For gel source data, see Supplementary Fig. 8. **c**, Expression changes in mouse TECs expressing the *Foxn4* gene from the cephalochordate *B. lanceolatum* (Bl). **d**, Evolutionary emergence of tolerogenic factors. Sequence

information for representative members of the *Aire* and *Fezf2* genes can be found in the following Genbank accession numbers. *Aire*: ADZ48462 (*Mus musculus*); XP_043558858 (*Chiloscyllium plagiosum*). *Fezf1*: XP_006505238 (*Mus musculus*); XP_043564758 (*Chiloscyllium plagiosum*); XP_032819022 (Fezf1-like, *Petromyzon marinus*); XP_061423322 (Fezf1-like, *Lenthenteron reissneri*); XP_039251141 (Fezf1-like, *Styela clava*); HM245959 (Fezf1-like, *Branchiostoma lanceolatum*). *Fezf2*: XP_030103750 (*Mus musculus*); XP_043564156 (*Chiloscyllium plagiosum*); XP_032822733 (Fezf2-like, *Petromyzon marinus*); XP_061407634 (Fezf2-like, *Lethenteron reissneri*).

# Reporting Summary

## Statistics

For all statistical analyses, confirm that the following items are present in the figure legend, table legend, main text, or Methods section.

| n/a | Confirmed | |
|---|---|---|
| ☐ | ☒ | The exact sample size (*n*) for each experimental group/condition, given as a discrete number and unit of measurement |
| ☐ | ☒ | A statement on whether measurements were taken from distinct samples or whether the same sample was measured repeatedly |
| ☐ | ☒ | The statistical test(s) used AND whether they are one- or two-sided<br>*Only common tests should be described solely by name; describe more complex techniques in the Methods section.* |
| ☐ | ☒ | A description of all covariates tested |
| ☐ | ☒ | A description of any assumptions or corrections, such as tests of normality and adjustment for multiple comparisons |
| ☐ | ☒ | A full description of the statistical parameters including central tendency (e.g. means) or other basic estimates (e.g. regression coefficient) AND variation (e.g. standard deviation) or associated estimates of uncertainty (e.g. confidence intervals) |
| ☐ | ☒ | For null hypothesis testing, the test statistic (e.g. $F$, $t$, $r$) with confidence intervals, effect sizes, degrees of freedom and $P$ value noted<br>*Give P values as exact values whenever suitable.* |
| ☒ | ☐ | For Bayesian analysis, information on the choice of priors and Markov chain Monte Carlo settings |
| ☒ | ☐ | For hierarchical and complex designs, identification of the appropriate level for tests and full reporting of outcomes |
| ☒ | ☐ | Estimates of effect sizes (e.g. Cohen's *d*, Pearson's *r*), indicating how they were calculated |

*Our web collection on statistics for biologists contains articles on many of the points above.*

## Software and code

Policy information about availability of computer code

| Data collection | FACS Diva Software v8.0.2, Summit 5.5 (MoFlow) |
|---|---|
| Data analysis | FlowJo 9.3.1 for flow cytometric analyses; GraphPad Prism 9.5.1<br><br>Raw data and the code to generate the figures for this manuscript can be found on Github (https://github.com/osthomas/mimetics_evodevo).<br><br>Analyses were run in conda (23.3.1) / mamba (1.4.2) environments with the following specifications:<br><br>R: r-base=4.2.3, r-here=1.0.1, r-tidyverse=1.3.2, r-ggplot2=3.5.1, r-remotes=2.4.2, r-devtools=2.4.5, r-matrix=1.6_1, r-matrixstats=1.0.0, r-patchwork=1.2.0, r-geomtextpath=0.1.4, r-dbplyr<=2.3.4, r-desctools=0.99.51, r-writexl=1.5.0, bioconductor-biomart=2.54.0, bioconductor-aucell=1.20.1, bioconductor-scran=1.26.0, bioconductor-scater=1.26.0, bioconductor-batchelor=1.14.0, bioconductor-complexheatmap=2.14.0<br><br>cellbender: python=3.7; pip: cellbender=0.3.0<br><br>sccoda: python=3.12.4; pip: pertpy=0.7.0, jax=0.4.30<br><br>scrnaseq: python=3.11.9, r-base=4.2.3, r-reticulate=1.38.0, quarto=1.5.57, r-tidyverse=1.3.2, r-ggplot2=3.5.1, r-remotes=2.4.2, r-devtools=2.4.5, r-patchwork=1.2.0, r-geomtextpath=0.1.4, |

```
r-matrix=1.6_1, r-matrixstats=1.0.0, r-hdf5r=1.3.10, bioconductor-aucell=1.20.1,
bioconductor-scran=1.26.0, bioconductor-scater=1.26.0, bioconductor-batchelor=1.14.0,
bioconductor-scdblfinder=1.12.0, bioconductor-complexheatmap=2.14.0,
bioconductor-zellkonverter=1.8.0, anndata=0.11.1, scanpy=1.10.3, cellrank=2.0.6; pip: igraph=0.11.8,
leidenalg=0.10.2

zellkonverter: r-base=4.2.3, r-tidyverse=1.3.2, bioconductor-scran=1.26.0,
bioconductor-scater=1.26.0, bioconductor-zellkonverter=1.8.0
```

For manuscripts utilizing custom algorithms or software that are central to the research but not yet described in published literature, software must be made available to editors and reviewers. We strongly encourage code deposition in a community repository (e.g. GitHub). See the Nature Portfolio guidelines for submitting code & software for further information.

# Data

Policy information about availability of data

All manuscripts must include a data availability statement. This statement should provide the following information, where applicable:

- Accession codes, unique identifiers, or web links for publicly available datasets
- A description of any restrictions on data availability
- For clinical datasets or third party data, please ensure that the statement adheres to our policy

Primary read files and expression count files for the single cell RNA sequencing datasets were reported previously (Nusser A et al. Developmental dynamics of two bipotent thymic epithelial progenitor types. Nature 606, 165 (2022)) and are available from GEO (accession GSE106856). No restrictions apply.
Primary read files and expression count files for the bulk RNA sequencing datasets are available from GEO (accession numbers GSE272144; GSE272064; GSE272063). No restrictions apply.
Primary read files and expression count files for the single cell RNA sequencing datasets were reported previously (Nusser A et al. Developmental dynamics of two bipotent thymic epithelial progenitor types. Nature 606, 165 (2022)) and are available from GEO (accession GSE106856). No restrictions apply.
Primary read files and expression count files for the bulk RNA sequencing datasets are available from GEO (accession numbers GSE272144; GSE272064; GSE272063). No restrictions apply.
Primary read files and processed files for the single nuclear RNA sequencing datasets are available from GEO (accession GSE288957). No restrictions apply.
Background signature gene sets are available from PanglaoDB (https://panglaodb.se/markers.html), the Tabula Muris repository (https://github.com/czbiohub-sf/tabula-muris/tree/dedd8352d4348150e199162f966f7442976acdd3/22_markers) and MSigDB (https://www.gsea-msigdb.org/gsea/msigdb/mouse/genesets.jsp?collection=M8).

# Research involving human participants, their data, or biological material

Policy information about studies with human participants or human data. See also policy information about sex, gender (identity/presentation), and sexual orientation and race, ethnicity and racism.

| | |
|---|---|
| Reporting on sex and gender | n/a |
| Reporting on race, ethnicity, or other socially relevant groupings | n/a |
| Population characteristics | n/a |
| Recruitment | n/a |
| Ethics oversight | n/a |

Note that full information on the approval of the study protocol must also be provided in the manuscript.

# Field-specific reporting

Please select the one below that is the best fit for your research. If you are not sure, read the appropriate sections before making your selection.

☒ Life sciences ☐ Behavioural & social sciences ☐ Ecological, evolutionary & environmental sciences

For a reference copy of the document with all sections, see nature.com/documents/nr-reporting-summary-flat.pdf

# Life sciences study design

All studies must disclose on these points even when the disclosure is negative.

| Sample size | The sample sizes for each experiment are indicated in the figure legends. Sample sizes were based on our experience and accepted practice in the respective fields, balancing statistical robustness, resource availability and animal welfare. No statistical methods were used to predetermine sample size. |
|---|---|
| Data exclusions | No data were excluded |
| Replication | The sample sizes for each experiment are indicated in the figures or figure legends. RNAseq analysis was performed in a group-wise fashion, each group comprising at least two biological replicates; the group-wise comparison was not replicated. The results of several independent types of analyses are in agreement. |
| Randomization | Provided the transgenic status, and age of mice matched the experimental requirements, mice were randomly assigned to experimental groups. |
| Blinding | Blinding was not possible because the thymus phenotype, ie. the transgenic status of the respective mouse, is evident from flow cytometry, imaging analysis, or genotyping information. |

# Reporting for specific materials, systems and methods

We require information from authors about some types of materials, experimental systems and methods used in many studies. Here, indicate whether each material, system or method listed is relevant to your study. If you are not sure if a list item applies to your research, read the appropriate section before selecting a response.

## Materials & experimental systems

| n/a | Involved in the study |
|---|---|
| ☐ | ☒ Antibodies |
| ☒ | ☐ Eukaryotic cell lines |
| ☒ | ☐ Palaeontology and archaeology |
| ☐ | ☒ Animals and other organisms |
| ☒ | ☐ Clinical data |
| ☒ | ☐ Dual use research of concern |
| ☒ | ☐ Plants |

## Methods

| n/a | Involved in the study |
|---|---|
| ☒ | ☐ ChIP-seq |
| ☐ | ☒ Flow cytometry |
| ☒ | ☐ MRI-based neuroimaging |

## Antibodies

| Antibodies used | Flow Cytometry:<br>Anti-EpCAM, host: Rat IgG2a, κ , conjugation: APC, clone: G8.8, supplier: BioLegend, cat#: 118214, dilution: 1:1000<br>Anti-EpCAM, host: Rat IgG2a, κ, conjugation: Biotin, clone: G8.8, supplier: BioLegend, cat#: 118204, dilution: 1:1000<br>Anti-CD45, host: Rat IgG2b, κ, conjugation: PE/Cy7, clone: 30-F11, supplier: BioLegend, cat#: 103114, dilution: 1:2000<br>Anti-Ly51 (alias: BP-1), host: Rat / IgG2a, kappa, conjugation: PE, clone: 6C3, supplier: ThermoFisher/eBioscience, cat#: 12-5891-82, dilution: 1:1600<br>UEA1, host: N/A, conjugation: FITC, clone: -, supplier: VectorLabs, cat#: FL-1061 , dilution: 1:1000<br>UEA1, host: N/A, conjugation: Biotin, clone: -, supplier: VectorLabs, cat#: B-1065-2 , dilution: 1:600<br>Anti-CD4, host: Rat IgG2b, κ , conjugation: FITC, clone: GK1.5, supplier: BioLegend, cat#: 100406, dilution: 1:1000<br>Anti-CD8a, host: Rat / IgG2a, kappa , conjugation: APC, clone: 53-6.7, supplier: ThermoFisher/eBioscience, cat#: 17-0081-82, dilution: 1:800<br>Anti-CD19, host: Rat / IgG2a, kappa, conjugation: PerCP/Cy5.5, clone: eBio1D3, supplier: ThermoFisher/eBioscience, cat#: 45-0193-82, dilution: 1:500<br>Anti-CD19, host: Rat / IgG2a, kappa, conjugation: PE/Cy7, clone: eBio1D3, supplier: ThermoFisher/eBioscience, cat#: 25-0193-82, dilution: 1:1000<br>Anti-CD45R (B220), host: Rat / IgG2a, kappa, conjugation: Biotin, clone: RA3-6B2, supplier: ThermoFisher/eBioscience, cat#: 13-0452-82, dilution: 1:200<br>Anti-IgM, host: Rat / IgG2a, kappa, conjugation: PE, clone: II/41, supplier: ThermoFisher/eBioscience, cat#: 12-5790-82, dilution: 1:300<br>Anti-CD93, host: Rat / IgG2b, kappa, conjugation: APC, clone: AA4.1, supplier: ThermoFisher/eBioscience, cat#: 17-5892-81, dilution: 1:300<br>anti-TCRb, host: Armenian hamster / IgG, conjugation: PE, clone: H57-597, supplier: ThermoFisher/eBioscience, cat#: 12-5961-82, dilution: 1:400<br>streptavidin, host: N/A, conjugation: FITC, clone: -, supplier: ThermoFisher/eBioscience, cat#: 11-4317-87, dilution: 1:1000 |
|---|---|

streptavidin, host: N/A, conjugation: eFluor450, clone: -, supplier: ThermoFisher/eBioscience, cat#: 48-4317-82, dilution: 1:1000
Anti-CD45, host: rat IgG2b, conjugation: MicroBeads, clone: 30-F11.1, supplier: Miltenyi Biotec, cat#: 130-052-301, dilution: 1:20, max. 1x10e8 cells/ml
Anti-Ter-119, host: rat IgG2b, conjugation: MicroBeads, clone: -, supplier: Miltenyi Biotec, cat#: 130-049-901, dilution: 1:20, max. 1x10e8 cells/ml

ISH:
Anti-Digoxigenin, host: sheep IgG, conjugation: alkaline phosphatase (AP), clone: polyclonal (Fab fragments), supplier: Roche, cat#: 11093274910, dilution: 1:2000
Anti-Digoxigenin, host: sheep IgG, conjugation: horseradish peroxidase (POD), clone: polyclonal (Fab fragments), supplier: Roche, cat#: 11207733910, dilution: 1:300
Anti-Fluorescein, host: sheep IgG, conjugation: horseradish peroxidase (POD), clone: , supplier: Roche, cat#: 11426346910, dilution: 1:300

Validation

All antibodies used in this study were sourced from commercial suppliers. Details about their validation strategies are available from:

BioLegend: https://www.biolegend.com/en-us/quality/product-development
ThermoFisher:
https://www.thermofisher.com/de/en/home/life-science/antibodies/invitrogen-antibody-validation.html
VectorLabs: https://vectorlabs.com/browse/antibodies/
Miltenyi Biotec: https://www.miltenyibiotec.com/US-en/products/macs-antibodies/antibody-validation.html

All antibodies were suitable for the applications as used in this study according to the manufacturers.

Anti-EpCAM (APC):
FC - Quality tested
Product Information: https://www.biolegend.com/ja-jp/products/apc-anti-mouse-cd326-ep-cam-antibody-4974

Anti-EpCAM (Biotin):
FC - Quality tested
Product Information:
https://www.biolegend.com/fr-ch/products/biotin-anti-mouse-cd326-ep-cam-antibody-4725

Anti-CD45:
FC - Quality tested
Product Information: https://www.biolegend.com/en-us/products/pe-cyanine7-anti-mouse-cd45-antibody-1903

Anti-Ly51 (alias: BP-1):
The 6C3 antibody has been tested by flow cytometric analysis of mouse bone marrow cells.
Product Information:
https://www.thermofisher.com/antibody/product/CD249-BP-1-Antibody-clone-6C3-Monoclonal/12-5891-82

UEA1:
Applications: Immunofluorescence, Glycobiology
Product Information: https://vectorlabs.com/products/fluorescein-ulex-europaeus-agglutinin/

UEA1 (Biotin):
Applications: Immunohistochemistry / Immunocytochemistry, Immunofluorescence, Blotting Applications, Elispot, ELISAs, Glycobiology
Product Information: https://vectorlabs.com/products/biotinylated-ulex-europaeus-agglutinin/

Anti-CD4:
FC - Quality tested
Product Information: https://www.biolegend.com/fr-ch/products/fitc-anti-mouse-cd4-antibody-248

Anti-CD8a:
Applications Tested: The 53-6.7 antibody has been tested by flow cytometric analysis of mouse thymocyte or splenocyte suspensions.
Product Information:
https://www.thermofisher.com/antibody/product/CD8a-Antibody-clone-53-6-7-Monoclonal/17-0081-82

Anti-CD19 (PerCP/Cy5.5):
Applications Tested: This eBio1D3 (1D3) antibody has been tested by flow cytometric analysis of mouse splenocytes.
Product Information:
https://www.thermofisher.com/antibody/product/CD19-Antibody-clone-eBio1D3-1D3-Monoclonal/45-0193-82

Anti-CD19 (PE/Cy7):
Applications Tested: This eBio1D3 (1D3) antibody has been tested by flow cytometric analysis of mouse splenocytes.
Product Information:
https://www.thermofisher.com/antibody/product/CD19-Antibody-clone-eBio1D3-1D3-Monoclonal/25-0193-82

Anti-CD45R:
Applications Tested: The RA3-6B2 antibody has been tested by flow cytometric analysis of mouse splenocytes.
Product Information:
https://www.thermofisher.com/antibody/product/CD45R-B220-Antibody-clone-RA3-6B2-Monoclonal/13-0452-82

Anti-IgM:
Applications Tested: This II/41 antibody has been tested by flow cytometric analysis of mouse bone marrow cells.
Product Information:
https://www.thermofisher.com/antibody/product/IgM-Antibody-clone-II-41-Monoclonal/12-5790-82

Anti-CD93:
Applications Tested: The AA4.1 antibody has been tested by flow cytometric analysis of mouse bone marrow and splenocyte cells.
Product Information:
https://www.thermofisher.com/antibody/product/CD93-AA4-1-Antibody-clone-AA4-1-Monoclonal/17-5892-81

anti-TCRb:
Applications Tested: The H57-597 antibody has been tested by flow cytometric analysis of mouse thymocytes and splenocytes.
Product Information:
https://www.thermofisher.com/antibody/product/TCR-beta-Antibody-clone-H57-597-Monoclonal/12-5961-82

streptavidin (FITC):
Reported Application: Flow Cytometric Analysis, Immunocytochemistry, Immunohistochemical Staining of Frozen Tissue Sections
Product Information:
https://www.thermofisher.com/order/catalog/product/11-4317-87?SID=srch-srp-11-4317-87

streptavidin (eFluor450):
Reported Application: Flow Cytometric Analysis
Product Information:
https://www.thermofisher.com/order/catalog/product/48-4317-82?SID=srch-srp-48-4317-82

Anti-CD45:
Mouse CD45 MicroBeads were developed for the positive selection or depletion of leukocytes from lymphoid and non-lymphoid tissues.
Product Information:
https://www.miltenyibiotec.com/DE-en/products/cd45-microbeads-mouse.html#130-052-301

Anti-Ter-119:
Anti-Ter-119 MicroBeads are suitable for positive selection or depletion of mouse erythrocytes or erythroid progenitors from lymphoid tissues.
Product Information:
https://www.miltenyibiotec.com/DE-en/products/anti-ter-119-microbeads-mouse.html#130-049-901

Anti-Digoxigenin:
Suitable for histochemistry according to product documentation
Product Information:
https://www.sigmaaldrich.com/FR/fr/product/roche/11093274910?srsltid=AfmBOop9KqUAP9_sx1EBxC88b93gHrQdb34DUlVaQut61-u7VivAvfpv

Anti-Digoxigenin:
Suitable for histochemistry according to product documentation
Product Information:
https://www.sigmaaldrich.com/FR/fr/product/roche/11207733910?srsltid=AfmBOood4LuUNBU3J7AkCs1QOOe1d56p61vcvZ7kpssbSI_P-MEdWXNa

Anti-Fluorescein:
Suitable for histochemistry according to product documentation
Product Information:
https://www.sigmaaldrich.com/FR/fr/product/roche/11426346910?srsltid=AfmBOoqgCTmub8Nik3Uci19GbkM4OWdKNWyr9aiBVgKJtncmjm7CDBo5

# Animals and other research organisms

Policy information about studies involving animals; ARRIVE guidelines recommended for reporting animal research, and Sex and Gender in Research

| Laboratory animals | C57BL/6 mice were maintained in the Max Planck Institute of Immunobiology and Epigenetics. Foxn1-/-, Foxn1:Cre, Rosa26-LSL-EYFP, Foxn1:mCardinal, Ascl1fl/fl, Foxn1:Bl_Foxn4, Foxn1:Cm_Foxn4, and |

Foxn1:Cm_Foxn1 as well as Foxn1:Fgf7 transgenic mice have been described previously. The Foxn1:Bmp4 transgene was created by Thomas Schlake and B.K. by inserting a cDNA fragment corresponding to nucleotides 497–1729 in GenBank accession number NM_007554.3 as a NotI fragment into pAHB1434. The Δ3ex2 Foxn1 deletion mutant (internal designation Chi6) transgene was generated by deletion of nucleotides 504-728 of mouse Foxn1 cDNA (Genbank accession number NM_008238.2) and insertion as a NotI fragment into pAHB1434. To generate transgenic mice, constructs were linearized and injected into FVB pronuclei according to standard protocols. The Foxn1Δ3ex2 mice were bred to a Foxn1-deficient background. Genotyping information is summarized in Supplementary Table 2. Mice carrying the original nu mutation (CByJ.Cg-Foxn1nu/J) were purchased from Charles River and used for snRNAsq experiments.
The zebrafish line carrying an internal deletion of the foxn1 gene was described.

Mice were analysed at the age of 4-6 weeks, unless otherwise stated.

Adult zebrafish (3 months of age) were used for experiments.

| | |
|---|---|
| Wild animals | No wild animals were used. |
| Reporting on sex | Phenotypes did not vary according to sex; for zebrafish embryos, sex determination is not possible |
| Field-collected samples | Ammocoete larvae of Lampetra planeri were caught from the field in the Freiburg region (Riedgraben, March-Neuershausen) (no precise age can be determined; body length 8-10 cm); juvenile Scyliorhiunus canicula specimens (directly after hatching) were kindly supplied by Markéta Kauka (Max Planck Institute for Evolutionary Biology, Plön, Germany); jjuvenile bamboo sharks (Chiloscyllium punctatum) were purchased from a local pet shop. Upon arrival at the laboratory, lampreys and sharks were euthanized using 0.02% tricaine methanesulfonate. |
| Ethics oversight | All animal experiments were performed in accordance with the relevant guidelines and regulations, approved by the review committee of the Max Planck Institute of Immunobiology and Epigenetics and the Regierungspräsidium Freiburg, Germany (mice: licenses 35-9185.81/G-12/85; 35-9185.81/G-16/67; zebrafish: license 35–9185.81/G-14/41). Ammocoete larvae of Lampetra planeri were caught from the wild in the Freiburg region (Riedgraben, March-Neuershausen) under permission by the local governmental authority (LandratsamtBreisgau-Hochschwarzwald, license 420.1.13-2024-034414). |

Note that full information on the approval of the study protocol must also be provided in the manuscript.

# Plants

| | |
|---|---|
| Seed stocks | *Report on the source of all seed stocks or other plant material used. If applicable, state the seed stock centre and catalogue number. If plant specimens were collected from the field, describe the collection location, date and sampling procedures.* |
| Novel plant genotypes | *Describe the methods by which all novel plant genotypes were produced. This includes those generated by transgenic approaches, gene editing, chemical/radiation-based mutagenesis and hybridization. For transgenic lines, describe the transformation method, the number of independent lines analyzed and the generation upon which experiments were performed. For gene-edited lines, describe the editor used, the endogenous sequence targeted for editing, the targeting guide RNA sequence (if applicable) and how the editor was applied.* |
| Authentication | *Describe any authentication procedures for each seed stock used or novel genotype generated. Describe any experiments used to assess the effect of a mutation and, where applicable, how potential secondary effects (e.g. second site T-DNA insertions, mosiacism, off-target gene editing) were examined.* |

# Flow Cytometry

## Plots

Confirm that:

☒ The axis labels state the marker and fluorochrome used (e.g. CD4-FITC).

☒ The axis scales are clearly visible. Include numbers along axes only for bottom left plot of group (a 'group' is an analysis of identical markers).

☒ All plots are contour plots with outliers or pseudocolor plots.

☒ A numerical value for number of cells or percentage (with statistics) is provided.

## Methodology

| | |
|---|---|
| Sample preparation | Thymic epithelial cells have the surface phenotype EpCAM+/CD45−; thus, cell surface staining was performed using anti-EpCAM (G8.8), conjugated with APC (1:1000, BioLegend) or anti-EpCAM (G8.8), conjugated with biotin (1:1000, BioLegend), in combination with streptavidin, conjugated with eFluor 450 (1:1000, eBioscience), and anti-CD45 (30-F11), conjugated with PE Cy7 (1:2000, BioLegend) at 4°C in PBS supplemented with 0.5% BSA and 0.02% NaN3. In order to differentiate cells with past and acute Foxn1 expression, triple-transgenic Foxn1:Cre; RosaR26SLSYFP; Foxn1:mCardinal mice were used for cell sorting. Cells with past expression of Foxn1 were sorted as EpCAM+YFP+ mCardinal- cells, whereas cells with acute Foxn1 expression were sorted as EpCAM+YFP+ mCardinal+ cells. CD45−EpCAM+ cells [after negative enrichment using anti-CD45 magnetic-activated cell sorting (MACS) beads and |

anti–Ter-119 MACS beads, Miltenyi Biotec] were sorted directly into TRI reagent (T9424, Sigma-Aldrich). Cell sorting was carried out using the MoFlow instrument (Dako Cytomation-Beckman Coulter) controlled with the Summit (5.5) software. Analytical flow cytometry was performed for TECs as follows: anti-EpCAM (G8.8), conjugated with APC (1:1000, BioLegend); anti-Ly51 (alias BP-1; 6C3), conjugated with PE (1:1600, eBioscience); UEA1, conjugated with FITC (1:1000, Vector Labs) or UEA1, conjugated with biotin (1:600, Vector Labs), in combination with streptavidin, conjugated with eFluor 450 (1:1000, eBioscience). When analysis of haematopoietic fractions was desired, thymocyte suspensions were prepared in parallel by mechanical liberation, best achieved by gently pressing thymic lobes through 40 μm sieves. Cell surface staining [anti-CD45 (30-F11), conjugated with PE/Cy7 (1:2000, BioLegend); anti-CD4 (GK1.5), conjugated with FITC (1:1000, BioLegend); anti-CD8a (53-6.7), conjugated with APC (1:800, eBioscience); anti-TCRβ (H57-597), conjugated with PE (1:400, eBioscience); anti-CD19 (eBio1D3), conjugated with PerCP/Cy5.5 (1:500, eBioscience) or PE/Cy7 (1:1000, eBioscience); anti-B220 (alias CD45R; RA3-6B2), conjugated with biotin (1:200, eBioscience); anti-IgM (II/4.1), conjugated with PE (1:300, eBioscience), anti-CD93 (alias C1qRp; AA4.1), conjugated with APC (1:300, eBioscience); streptavidin conjugated with eFluor 450 or FITC (1:1000, eBioscience)] was performed at 4°C in PBS supplemented with 0.5% BSA and 0.02% NaN3. Flow cytometry experiments were evaluated using FACSDiva (8.0.2) and FlowJo (9.3.1) software. The relevant gating strategies are shown in Supplementary Figure 4.

| | |
|---|---|
| Instrument | BD Fortessa II; MoFlow; both from Dako Cytomation-Beckman Coulter |
| Software | FACS Diva Software v8.0.2, Summit 5.5 (MoFlow), FlowJo 9.3.1 for flow cytometric analyses<br>Data analysis was carried out in conda environments |
| Cell population abundance | Purity was determined by running a purity check of the sorted populations after the sort was completed. |
| Gating strategy | All samples were initially gated using forward and side scatter to identify events corresponding to cells, doublets are excluded by gating on single cells using forward scatter height vs. area, alive cells were selected by negativity for the viability dye Fluoro Gold, the follow gating steps are according to the marker genes described in the manuscript |

☒ Tick this box to confirm that a figure exemplifying the gating strategy is provided in the Supplementary Information.

