## [Peer Review file · Nature]

Developmental trajectory and evolutionary origin of thymic mimetic cells

Corresponding Author: Dr Thomas Boehm

This file contains all reviewer reports in order by version, followed by all author rebuttals in order by version. Parts of this Peer Review File have been redacted as indicated to maintain the confidentiality of unpublished data.

Version 0:

Reviewer comments:

Referee #1

(Remarks to the Author)

The manuscript by Nusser et al. addresses the developmental origins of recently identified thymic mimetic cells. The authors used a combination of genetic, lineage tracing, and evolutionary approaches to identify the embryonic and postnatal timepoints at which different mimetic cell populations emerge. The authors identify distinct waves of mimetic cells as well as contributions of Foxn1 to shaping mimetic cell heterogeneity. An interesting concept proposed by the authors is that the emergence of particular mimetic cells occurred in parallel and perhaps in response to the development of their 'linked' organs or tissues.

The developmental origins and differentiation pathways for mimetic cells have become important areas of investigation since their discovery. This fascinating and elegant study goes some way to addressing this. While the study establishes that distinct mimetic cell types emerge at different time points during thymic development and are differentially dependent on particular transcription factors and signaling molecules, additional experiments are required to provide a more detailed, high-resolution depiction of the progenitors and differentiation pathways that give rise to individual mimetic cell types.

Major comments

A striking finding from the analyses in Figs. 1a, b, and d is the enrichment of muscle mimetic cells at E15.5 or P1, suggesting that these cells emerge earlier in thymic development from early progenitors. In addition, the authors show that muscle cells develop in the absence of Foxn1. These findings are in line with the earlier observation from Michelson et al. (Cell 2022) that muscle mimetic cells were not enriched in cells with a history of Aire expression, suggesting a distinct developmental pathway for muscle mimetic cells. However, the developmental origin of muscle mimetic cells is not addressed by cellular barcoding experiments (Fig. 1e) due to the limited number of muscle cells. A more detailed analysis of this differentiation pathway could provide insights into the developmental origins of mimetic cells.

The cellular barcoding (Fig. 1e) identified a predominance of shared barcodes between postnatal progenitors, Aire+ mTECs and mimetic cells, in keeping with previous lineage tracing experiments demonstrating the 'post-Aire' nature of mimetics. However, there are some barcodes which were present in Aire+ mTECs but not mimetic cells e.g. 102, 91, 86 (P28 #3). Do the Aire+ cells with mimetic potential have distinct transcriptional phenotypes from those without?

Similarly, to what extent do postnatal progenitors or Aire+ mTECs have multipotent mimetic cell potential for the groups of mimetic cells identified in Fig. 3i? A more detailed analysis of the cellular barcoding of TECs in Fig. 1e (or, if required, a larger dataset), would provide more insight into the differentiation pathways for mimetics. For example, in instances where barcodes are shared across mimetic postnatal progenitors/Aire+ mTECs/mimetics, are barcodes shared across particular classes of mimetic cells? Where does the bifurcation in mimetic cell differentiation potential occur?

Referee #2

(Remarks to the Author)

This is a MS from Nusser et al. that explores the ontology of the development of thymic mimetic cell populations in vertebrates. There has been a recent appreciation for the thymus to have medullary TEC populations that mimic or have

similarity to peripheral cells and these cells include keratinocytes, tuft cells, respiratory epithelial cells, enteroendocrine cells, and goblet cells that have now been firmly described in both mouse and humans (Ref #2). In this study, the investigators attempt to dissect the origins of mimetic cells and their development. They first perform RNA-Seq experiments on prenatal, new born and day 28 mice and find an organized development of mimetic cells that unfolds with increasing age that can be separated into two waves (keratinocyte and tuft cells later). They also show that the majority of these cells developed from a Foxn1 expressing precursor cell using a sophisticated bar coding method in mouse. The group then moves into using Foxn1 mutant and replacement approaches to further dissect the mimetic cell development. Of note, in the thymic rudiment of Foxn1 ^{-/-} mice there is evidence of mimetic cell gene expression of respiratory epithelial markers (Foxj1) . The authors then turned to manipulation of TEC's via transgenic introduction of Bmp4 and Fgf7 which increases early progenitors and by RNA seq analysis they find shifts in mimetic cell types. Next they turn to a unique mouse model where the Foxn1 gene is replaced by a delta 3 exon 2 allele with hypomorphic Foxn1 activity. Again they find enrichment for early progenitors and a shift in mimetic types that are altered (less microfold, skin, etc). In final set of experiments the team looks into a FoxN1 replacement mouse system where mouse Foxn1 is replaced by primordial alleles of Foxn1 from the Foxn4 genes of older vertebrates and here they generally find that older alleles cannot support full mTEC maturation or the development of most mimetic populations.

This paper has a series of interesting findings about the development of mimetic populations in these various mutant alleles. At the same time there are several concerns here that dampen enthusiasm for novelty of the findings. First, there is the technical concern that the majority of new data in this paper is bulk RNA-Seq from TEC populations rather than single cell RNA-Seq. This leads to a key problem in that how does one know that the respiratory/goblet mimetic cells observed in the Foxn1 null mutants or in the Foxn1 replacement models with Foxn4 alleles are truly the same population as those observed in WT mice? Indeed, evidence already points to these cells passing through an Aire+ lineage in the WT situation (Michelson et al. Cell 2022) and thus the simple conclusion that they do not rely on Foxn1 or Aire may be incorrect. This level of robust analysis is absent from the paper. Likewise, it has long been known that Foxn1 null mice harbor a respiratory like cell in the thymic rudiment (J Immunol 2005 Oct 1;175(7):4331-7) but it is not clear if this is the same cell that is described in WT mTEC cells.

Second, the majority of the findings here may be more in line with a model whereby Foxn1 helps promote a mature mTEC lineage that can then give rise to most mimetic populations. The direct evidence that there was evolution of Foxn1 to give rise to mimetic populations like liver is not convincing here. One would have to put mouse Foxn1 into an older organism and show this rescued the production of liver like mimetic cells to reach such a conclusion rather than reverse which is in the current ms.

Finally, most of the mimetic findings here could be indirect sequelae of the manipulation of Foxn1 rather than direct for promoting mimetic cell differentiation. Hypomorphic Foxn1 could just lead to a TEC population that can't fully differentiate to a mature mTEC that then gives rise to most mimetic populations in the thymus.

Minor comment:

Ascl1 as a factor in the TEC development and mimetic cells was already recently described with similar conclusions (Nature 2023 Oct;622(7981):164-172).

Version 1:

Reviewer comments:

Referee #1

(Remarks to the Author)

The authors have provided additional data to address the points raised, conducting both RNA in situ hybridization to address spatial relationships between thymic progenitors and particular mimetic cell subtypes, and a recently developed computational approach to identify putative developmental relationships. The results of both analyses support the manuscript's conclusion – that different mimetic cell types have distinct origins – but fall short of definitively identifying individual mimetic cell precursors and the signals that promote their distinct developmental trajectories. I understand the challenges associated with profiling large numbers of mimetic cells; however a larger single-cell multiomic analysis of TECs may yield further insights.

Referee #2

(Remarks to the Author)

The text and figures have been thoughtfully revised, including the addition of new data responsive to reviewer comments. This new data strengthens some of the claims made, for example by the addition of snRNA-seq data to Figure 4. Overall, the revised study remains conceptually interesting and thought provoking and is likely to engage a broad audience due to its evolutionary bent on a timely subject and the intersection of immunology and developmental biology. However, because of its breadth and ambition, many of the conclusions remain quite speculative, with multiple possible interpretations to the data presented.

We thank the referees for the time they have spent on the review and their constructive comments, which have allowed us to clarify a number of aspects and to considerably strengthen our manuscript by adding the results of several additional experiments, including some that addressed the potential presence of cells expressing peripheral antigens in the thymoid of lampreys. Our responses to the individual comments are indicated in blue font.

Referee #1 (Remarks to the Author):

The manuscript by Nusser et al. addresses the developmental origins of recently identified thymic mimetic cells. The authors used a combination of genetic, lineage tracing, and evolutionary approaches to identify the embryonic and postnatal timepoints at which different mimetic cell populations emerge. The authors identify distinct waves of mimetic cells as well as contributions of Foxn1 to shaping mimetic cell heterogeneity. An interesting concept proposed by the authors is that the emergence of particular mimetic cells occurred in parallel and perhaps in response to the development of their ‘linked’ organs or tissues.

The developmental origins and differentiation pathways for mimetic cells have become important areas of investigation since their discovery. This fascinating and elegant study goes some way to addressing this. While the study establishes that distinct mimetic cell types emerge at different time points during thymic development and are differentially dependent on particular transcription factors and signaling molecules, additional experiments are required to provide a more detailed, high-resolution depiction of the progenitors and differentiation pathways that give rise to individual mimetic cell types.

Response:

We thank the reviewer for her/his encouraging comments on our work and her/his constructive critiques. In response, we have substantially expanded and refined our study to address the points raised by the reviewer as detailed below.

Major comments

A striking finding from the analyses in Figs. 1a, b, and d is the enrichment of muscle mimetic cells at E15.5 or P1, suggesting that these cells emerge earlier in thymic development from early progenitors. In addition, the authors show that muscle cells develop in the absence of Foxn1. These findings are in line with the earlier observation from Michelson et al. (Cell 2022) that muscle mimetic cells were not enriched in cells with a history of Aire expression, suggesting a distinct developmental pathway for muscle mimetic cells. However, the developmental origin of muscle mimetic cells is not addressed by cellular barcoding experiments (Fig. 1e) due to the limited number of muscle cells. A more detailed analysis of this differentiation pathway could provide insights into the developmental origins of mimetic cells.

Response:

In response to the reviewer's comments, we have expanded our analysis of the muscle mimetic cells in the thymic microenvironment by RNA in situ hybridization on thymus tissue sections.

To this end, we recorded the expression pattern of *Myog*, the muscle lineage-defining transcription factor in the thymus of newborn and 4 week-old mice. Two additional interesting features emerged from this experiment. (a) At P0, *Myog*-positive cells often occur in small but well-separated clusters, suggesting their multi-focal simultaneous emergence; at P28, the *Myog*-positive cells have dispersed among other cells in the medulla and no obvious clusters are detectable at this time point; this pattern is compatible with an early burst of muscle cell development. (b) Confirming the result of the original scRNA-seq analysis, *Myog*-positive cells exhibit a relative decrease of about three-fold between P0 and P28 (new Extended Data Figure 2a-d).

Furthermore, we have used a similar spatial analysis aimed at investigating the localization of mimetic cells relative to TEC precursor populations, and with respect to each other to learn more about the developmental trajectory of other mimetic cells. To this end, we have carried out several additional experiments.

We first determined the expression pattern of *Igfbp5*, a gene that we had assigned to a group of genes characterizing the postnatal progenitor (Nusser et al, 2022 [PMID:35614226]); in the meantime, this assignment was confirmed by others (Weiler et al., 2024 [PMID:38871986]) using their CellRank algorithm for cell fate analysis. RNA in situ hybridization shows that *Igfbp5* is expressed predominantly in the medullary region, but lacks co-expression with the *Aire*-expressing cell population, indicating their distinct identity. However, *Aire*-expressing cells are always situated close to *Igfbp5*-expressing cells, indicative of their close developmental relationship. The relative proportions of the postnatal progenitor and the *Aire*-stage populations from the scRNAseq analysis (Extended Data Figure 1c) and the RNA in situ experiments (*Igfbp5*-expressing cells vs. *Aire*-expressing cells [New Extended Data Figure 4]) are remarkably similar, both indicating a ratio of about 2/3, providing a cross-validation of these analytical strategies.

With respect to mimetic cells, we focused on the localization of ionocytes (marked by the expression of *Foxi1*) and ciliated cells (marked by the expression of *Foxj1*). Whereas *Foxi1*-expressing ionocytes are often situated close to *Igfbp5*-expressing cells, *Foxj1*-expressing cells are well separated from *Igfbp5*-expressing cells, indicating a less tight developmental relationship; accordingly, *Foxi1*- and *Foxj1*-expressing cells are never found adjacent to one another (New Extended Data Figure 4c).

Collectively, these additional experiments support our conclusion that individual mimetic cell types exhibit distinct developmental pathways.

The cellular barcoding (Fig. 1e) identified a predominance of shared barcodes between postnatal progenitors, *Aire*⁺ mTECs and mimetic cells, in keeping with previous lineage tracing experiments demonstrating the 'post-*Aire*' nature of mimetics. However, there are some barcodes which were present in *Aire*⁺ mTECs but not mimetic cells e.g. 102, 91, 86 (P28 #3). Do the *Aire*⁺ cells with mimetic potential have distinct transcriptional phenotypes from those without?

Response:

We thank the reviewer for this interesting suggestion. In response to this comment, we have compared the transcriptomes of Aire-stage cells which carry barcodes exclusive to Aire-stage cells ("Aire+") versus those carrying barcodes shared between Aire-stage and mimetic cells ("Aire+/mimetic+"). The results indicate their close transcriptional similarity, as indicated by their co-localization on the integrated UMAP (Reviewer Figure 1a).

SENTENCES REDACTED

FIGURE PANELS REDACTED

Reviewer Figure 1: Differential gene expression patterns distinguishing cells marked by "Aire+" and "Aire+/mimetic+" barcodes. (a) Integrated UMAP for TECs from E16.5, P0, and P28 time points (see Extended Data Figure 1c for individual UMAPs). Dark grey cells represent Aire-stage cells as defined in Extended Data Figure 1. Aire+ cells (identified by barcodes 2, 63 in E16.5 TECs; 5, 8, 56 in P0 TECs; 3/15, 19, 26 in P28#1; 8, 41, 45 in P28#2; 5, 6, 21, 84, 86, 87, 88, 89, 91, 93, 94/2, 95/7, 102 in P28#3) and "Aire+/mimetic+" cells (identified by barcodes 3/12, 13, 14, 109, 110, in P28#1; 43 in P28#2; 34/2, 68/2, 68/5, 83, 85, 91/5 in P28#3) are identified in black and orange colour, respectively.

SENTENCES REDACTED

Similarly, to what extent do postnatal progenitors or Aire+ mTECs have multipotent mimetic cell potential for the groups of mimetic cells identified in Fig. 3i? A more detailed analysis of the cellular barcoding of TECs in Fig. 1e (or, if required, a larger dataset), would provide more insight into the differentiation pathways for mimetics. For example, in instances where barcodes are shared across mimetic postnatal progenitors/Aire+ mTECs/mimetics, are

barcodes shared across particular classes of mimetic cells? Where does the bifurcation in mimetic cell differentiation potential occur?

Response:

The reviewer correctly observed that the resolution of our barcoding scheme lacks power to precisely identify the bifurcation of the individual mimetic cell groups, especially those that represent only a small fraction of TECs. For this reason, we have focused our study on using several genetic models to address the possibility of distinct developmental and genetic requirements for individual mimetic cell types. We have emphasized this strategic aspect more clearly in the revised version. Collectively, the different genetic constellations used in our study suggest that some cells develop at an early stage (muscle cells), others lack the requirement for *Foxn1* transcription factor function (such as ciliated cells) – and by inference cannot belong to the post-Aire lineage – (see the results of the new snRNAseq experiment comparing the transcriptomes of cells exhibiting the ciliated cell signature in *Foxn1*-sufficient and *Foxn1*-deficient thymic microenvironments [New Fig. 4c]), yet others, such as enterohepatic cells require *Foxn1* activity, and hence likely belong to the post-Aire compartment, as suggested previously (Michelson et al., 2022).

Encouraged by the correct identification of postnatal/mTEC progenitor cells by the CellRank algorithm (Weiler et al., 2024 [PMID:38871986]) we used this method to assess the developmental relationship of the numerically prominent mimetic cell types. To this end, we integrated all cells from E16.5, P0, and P28, chose a starting cell (in our case a cell of the early progenitor population) for calculation of diffusion pseudotime, and calculated fate probabilities based on pseudotime and overall transcriptional connectivity. The results of this analysis show that cells with fate probabilities towards the cTEC lineage may also contribute to the mTEC compartment; this outcome is compatible with our previous conclusion about the successive emergence and differentiation potential of embryonic and postnatal progenitor populations established by the barcode analysis (Nusser et al., 2022). The CellRank algorithm successfully recovered a number of previously identified mimetic types as terminal states of differentiation; remarkably, when grouped according to fate probability, ciliated cells are predicted to split off from the canonical developmental trajectory at an early time point (New Extended Data Figure 5; New Supplementary Figure 3). When forced to include all mimetic cell populations previously identified via signature scores, the algorithm returns a pattern indicative of an early separation of muscle cells, among others, from the early progenitor (Supplementary Figure 3). Collectively, the results obtained by analysis of different developmental time points, the composition of thymic microenvironments in different genetic models, and the computational fate predictions all support our conclusion of different types of mimetic cells. Of note, based on our results, future experiments might use an *Igfbp5*-driver to increase the resolution of the barcode analysis amongst the descendants of the postnatal progenitor population.

Referee #2 (Remarks to the Author):

This is a MS from Nusser et al. that explores the ontology of the development of thymic mimetic cell populations in vertebrates. There has been a recent appreciation for the thymus to have medullary TEC populations that mimic or have similarity to peripheral cells and these cells include keratinocytes, tuft cells, respiratory epithelial cells, enteroendocrine cells, and

goblet cells that have now been firmly described in both mouse and humans (Ref #2). In this study, the investigators attempt to dissect the origins of mimetic cells and their development. They first perform RNA-Seq experiments on prenatal, new born and day 28 mice and find an organized development of mimetic cells that unfolds with increasing age that can be separated into two waves (keratinocyte and tuft cells later). They also show that the majority of these cells developed from a Foxn1 expressing precursor cell using a sophisticated bar coding method in mouse. The group then moves into using Foxn1 mutant and replacement approaches to further dissect the mimetic cell development. Of note, in the thymic rudiment of Foxn1 $-/-$ mice there is evidence of mimetic cell gene expression of respiratory epithelial markers (Foxj1). The authors then turned to manipulation of TEC's via transgenic introduction of Bmp4 and Fgf7 which increases early progenitors and by RNA seq analysis they find shifts in mimetic cell types. Next they turn to a unique mouse model where the Foxn1 gene is replaced by a delta 3 exon 2 allele with hypomorphic Foxn1 activity. Again they find enrichment for early progenitors and a shift in mimetic types that are altered (less microfold, skin, etc). In final set of experiments the team looks into a FoxN1 replacement mouse system where mouse Foxn1 is replaced by primordial alleles of Foxn1 from the Foxn4 genes of older vertebrates and here they generally find that older alleles cannot support full mTEC maturation or the development of most mimetic populations.

This paper has a series of interesting findings about the development of mimetic populations in these various mutant alleles.

Response:

We thank the reviewer for her/his positive assessment of our work. In response, we have substantially expanded and refined our study to address the points raised by the reviewer as detailed below, with a particular focus on the evolutionary origin of mimetic cells.

At the same time there are several concerns here that dampen enthusiasm for novelty of the findings. First, there is the technical concern that the majority of new data in this paper is bulk RNA-Seq from TEC populations rather than single cell RNA-Seq. This leads to a key problem in that how does one know that the respiratory/goblet mimetic cells observed in the Foxn1 null mutants or in the Foxn1 replacement models with Foxn4 alleles are truly the same population as those observed in WT mice? Indeed, evidence already points to these cells passing through an Aire⁺ lineage in the WT situation (Michelson et al. Cell 2022) and thus the simple conclusion that they do not rely on Foxn1 or Aire may be incorrect. This level of robust analysis is absent from the paper. Likewise, it has long been known that Foxn1 null mice harbor a respiratory like cell in the thymic rudiment (J Immunol 2005 Oct 1;175(7):4331-7) but it is not clear if this is the same cell that is described in WT mTEC cells.

Response:

The reviewer correctly observed that our original manuscript did not include direct evidence of shared characteristics of mimetic cell candidates in the *Foxn1* $-/-$ rudiment versus the *Foxn1*-sufficient background. To address these concerns, we have performed additional experiments. New snRNAseq experiments were used to compare the transcriptomes of cells with a ciliated signature from *Foxn1*-sufficient and *Foxn1*-deficient genetic backgrounds. In this case, we have used a snRNAseq procedure rather than the method of scRNAseq of flow cytometrically purified mTECs in order to avoid potential problems associated with the enzymatic disruption of distinctly different tissue types. As expected, the *Foxn1*-sufficient

samples were unavoidably dominated by haematopoietic lineage cells, hence we focused our attention on mimetic cells that occur at relatively high frequency, especially on cells with a signature of ciliated cells, as requested by the reviewer. Cells with a pronounced Ciliated signature formed a single cluster, indicating that their overall transcriptomes are very similar and thus likely represent the same cell type. Importantly, cells from both the *Foxn1*-deficient and *Foxn1*-sufficient backgrounds were represented in this cluster at substantial proportions across all experimental replicates. We conclude that, although ciliated cells go through an initial *Foxn1*-positive stage, as evidenced by the presence of Foxn1-induced barcodes, they evidently do not require the activity of the Foxn1 transcription factor for their development.

We thank the reviewer for requesting this clarification which provided clear-cut evidence that mimetic cells differ in their genetic requirements for development.

Second, the majority of the findings here may be more in line with a model whereby Foxn1 helps promote a mature mTEC lineage that can then give rise to most mimetic populations.

Response:

The reviewer raises an important point. The analysis of ciliated cells described above confirms that Foxn1 activity is not an absolute requirement for mimetic cell development in general, although our results indicate that other cell types, such as those belonging to the enterohepatic lineage do depend, at least transiently, on Foxn1 activity, as also exemplified by our dual reporter clonal analysis (Fig. 2g,h). This finding represents one of the central conclusions of the present manuscript.

The direct evidence that there was evolution of Foxn1 to give rise to mimetic populations like liver is not convincing here. One would have to put mouse Foxn1 into an older organism and show this rescued the production of liver like mimetic cells to reach such a conclusion rather than reverse which is in the current ms.

Response:

In response to this interesting suggestion, we note the following. The Foxn1 transcription factor does not induce the formation of the thymic epithelium from undifferentiated pharyngeal progenitors; rather, it is required for the subsequent differentiation of the thymic microenvironment (see Nehls et al. 1996 [PMID:8629026]; Blackburn et al., 1996 [8650163]; Bleul et al., 2006 [16791198]). Hence, the experiment suggested by the reviewer is experimentally not feasible (at least at present), as it would require to additionally “engineer” the entire genetic network required to induce *Foxn1* expression into a non-vertebrate organism. Tunicates, the closest living relatives to vertebrates are not amenable to such complex genetic manipulations.

Nonetheless, we felt it important to address this valid concern by additional experiments, as follows. First, we have directly examined the presence of cells expressing key signature genes of mimetic cell types in the thymus of a cartilaginous fish (*S. canicula*), a representative of the most basal jawed vertebrates. The results shown in new Fig. 5, and New Extended Data Figure 10, indicate that such cells are located in the medulla of the thymus, often in small cell aggregates, exactly like the situation in the mouse. Because the thymic microenvironment of both cartilaginous and bony fishes expresses not only *Foxn1* but also *Foxn4* genes, we examined the thymopoietic capacities of these two paralogous genes in two ways. We directly compared the thymopoietic capacity of shark *FOXN4* and *FOXN1* genes in

our *Foxn1*-replacement model and assessed the activities of the two paralogs, individually and in combination. In a second experiment, we determined the presence of mimetic cells in the thymus of *foxn1*-deficient zebrafish. The results demonstrate a remarkable similarity between the contributions of zebrafish *foxn4* (as assessed by comparing the thymic phenotypes of *foxn1*^{+/+};*foxn4*^{+/+} and *foxn1*^{-/-};*foxn4*^{+/+} fish) and shark FOXN4 (as assessed by comparing the thymic phenotypes in the individual FOXN4 and FOXN1 reconstitutions) (new Fig. 5). This observation clearly indicates that evolutionarily “older” (that is, FOXN4) versions of the Foxn1/4 gene family have the capacity to support all known mimetic cell types, in qualitative terms, although distinct differences exist to Foxn1 genes.

In order to additionally strengthen our hypothesis that vertebrate-specific innovations, such as the liver, are accompanied by appropriate representations in the thymic microenvironment, we have turned our attention to the sister group of jawed vertebrates, the jawless fishes. Like their jawed vertebrate sister lineage, jawless vertebrates possess a liver. Furthermore, lampreys also possess a thymus equivalent, termed thymoid (Bajoghli et al., 2011 [2129372]). We have now studied the microanatomy of this tissue further with special reference to the potential presence of cells expressing peripheral tissue antigens. As shown in New Fig. 5, our results show that the lamprey thymoid indeed harbours putative mimetic cells in the inner area of the thymoid (which we tentatively refer to as medulla), whereas *CDAI*-expressing developing thymocytes are situated at the outer edge of the thymoid, akin to the RAG-expressing thymocytes in the cortex of the mammalian thymus. These findings strengthen our conclusion that the thymopoietic microenvironment of vertebrates in general provides a representation of peripheral self, even in a situation where the molecular identities of the antigen receptors radically differ (see Boehm et al., 2018 [29144837] for review).

We thank the reviewer for prompting us to assess the pan-vertebrate nature of self-representation in the thymus.

Finally, most of the mimetic findings here could be indirect sequelae of the manipulation of Foxn1 rather than direct for promoting mimetic cell differentiation. Hypomorphic Foxn1 could just lead to a TEC population that can't fully differentiate to a mature mTEC that then gives rise to most mimetic populations in the thymus.

Response:

The reviewer raises an interesting possibility. However, we think that this is an unlikely scenario for the following reasons. (1) In both shark FOXN4-driven and shark FOXN1-driven microenvironments (as revealed by the reconstructions), no autoimmune phenomena are observed, and the TECs exhibit all key aspects of TEC maturity, such as CD80 expression etc. (see also Swann et al., 2020 [33246964]). Hence, not only *Foxn1*, but also its paralogous companion *Foxn4* is capable of directing peripheral self-antigen representation in the thymus, even if differences exist in relative proportions among the mimetic cell subsets. (2) Furthermore, in the *foxn1*-deficient zebrafish mutant, in which thymus differentiation is entirely dependent on endogenous *foxn4* activity, no autoimmune phenomena are observed (see also Schorpp et al., 2023 [33724048]). Note also that in the mice in which TEC function is restored by expression of shark FOXN1 or FOXN4, the FOXN4-dependent contribution to the mimetic cell population is almost indistinguishable from that of the zebrafish *foxn1*^{-/-};*foxn4*^{+/+} constellation.

However, the reviewer is correct to note that we have investigated one constellation where we have deliberately used a hypomorphic *Foxn1* mutant, designated *Foxn1*^{Δ3ex2}. Remarkably,

this mouse mutant did not show any autoimmune phenomena either and was used here to assess the contribution of postnatal progenitors to the population of mimetic cells. Because of the time-delayed onset of postnatal progenitor differentiation in this genetic model, it allowed us to separate "early" from "late" contributions to mimetic cell pool in the adult thymus. Hence, we have no reason to think that the resulting phenotype is the result of a qualitatively impaired TEC differentiation.

Minor comment:

Ascl1 as a factor in the TEC development and mimetic cells was already recently described with similar conclusions (Nature 2023 Oct;622(7981):164-172).

Response:

We thank the reviewer for alerting us to this finding, which we have now properly acknowledged in the revised version. Of note, this concordance provides independent support for the choice of analysing mimetic cell differentiation using cell-type specific gene signatures extracted from bulk RNAseq data as an analytical strategy comparable to scRNAseq-type studies.